# Comparative Transcriptomics Reveal Differential Expression of Coding and Non-Coding RNAs in Clinical Strains of *Mycobacterium tuberculosis*

**DOI:** 10.3390/ijms26010217

**Published:** 2024-12-30

**Authors:** Nontobeko Eunice Mvubu, Divenita Govender, Manormoney Pillay

**Affiliations:** 1School of Laboratory Medicine and Medical Sciences, College of Health Sciences, University of KwaZulu-Natal, Durban 4000, South Africa; pillayc@ukzn.ac.za; 2School of Life Sciences, College of Agriculture, Engineering and Science, University of KwaZulu-Natal, Durban 4000, South Africa; divenitagovender@gmail.com

**Keywords:** tuberculosis, *M. tuberculosis*, transcriptomics, ncRNA, gene regulation, RNA Seq

## Abstract

Coding and non-coding RNAs (ncRNAs) are potential novel markers that can be exploited for TB diagnostics in the fight against *Mycobacterium tuberculosis*. The current study investigated the mechanisms of transcript regulation and ncRNA signatures through Total RNA Seq and small (smRNA) RNA Seq followed by Bioinformatics analysis in Beijing and F15/LAM4/KZN (KZN) clinical strains compared to the laboratory strain. Total RNA Seq revealed differential regulation of RNA transcripts in Beijing (n = 1095) and KZN (n = 856) strains compared to the laboratory H37Rv strain. The KZN vs. H37Rv coding transcripts uniquely enriched fatty acids, steroid degradation, fructose, and mannose metabolism as well as a bacterial secretion system. In contrast, Tuberculosis and biosynthesis of siderophores KEGG pathways were enriched by the Beijing vs. H37Rv-specific transcripts. Novel sense and antisense ncRNAs, as well as the expression of these transcripts, were observed, and these targeted RNA transcripts are involved in cell wall synthesis and bacterial metabolism in a strain-specific manner. RNA transcripts identified in the current study offer insights into gene regulation of transcripts involved in the growth and metabolism of the clinically relevant KZN and Beijing strains compared to the laboratory H37Rv strain and thus can be exploited in the fight against Tuberculosis.

## 1. Introduction

Tuberculosis (TB), a global health threat, is one of the 10 causes of death across the world, with an incidence rate of 133 new cases per 100 000 population and an estimated death toll of 1.3 million in 2023 [1]. The most widespread TB disease cases in South Africa were caused by the W/Beijing and the F15/LAM4/KZN (KZN) families of the East-Asian and Euro-American lineages, respectively, with the latter family being the most common in the KwaZulu-Natal province since the mid-2000s [2,3]. With the COVID-19 pandemic plaguing the globe, a 1.3 million drop in the number of newly diagnosed people has been reported [4]. This is a result of reduced access to TB treatment and testing facilities, causing a drastic increase in TB-related deaths [4]. Despite progress made in TB pathogenesis research, very limited studies have investigated the role of small RNAs (sRNAs) in clinical strains of *M. tuberculosis* and their contribution to the persistence of these pathogens in the human host [5].

The discovery of regulatory noncoding sRNAs in *M. tuberculosis* in the late 2000s has provided insight into gene regulation and changes in molecular mechanisms in response to external stimuli [6,7]. The Mcr7 was one of the first sRNAs to be classified in *M. tuberculosis*. This transcript forms part of the PhoPR two-component system, which is essential for virulence in *M. tuberculosis*. Secretory proteins of the Twin Arginine Translocation protein secretion complex that is modulated by tatC mRNA are significantly reduced in the presence of Mcr7, indicating the critical role of this sRNA within this complex [8]. The expression of the *Mycobacterium tuberculosis* complex (MTBC) exclusive MTS1338 sRNA has been identified to facilitate adaptation and survival of the bacterium within the low pH conditions in the macrophage environment. Furthermore, MTS1338 overexpression triggered numerous transcriptome changes, such as the reduction in the translational activity and decreased growth, indicative of the bacterium transitioning to a dormant state [9]. The mycobacterial regulatory sRNA in iron (Mrsl) directly binds to the target, *bfrA*, a response activated during iron-limiting conditions resulting in iron-sparing, thus allowing for the survival of the *Mycobacterium*. This is activated as an anticipatory response toward iron deprivation that is typical within a macrophage environment [10]. These are just a few of the small transcripts identified in MTBC, which mainly target several mRNAs, in an antisense manner, and are activated as a stress, virulence, pathogenicity, and adaptive response [11,12]. The majority of the over 200 sRNAs identified within the MTBC, were predicted through in silico tools, while few have been confirmed and functionally characterized [13].

Although the genomic sequences are highly similar within the MTBC, even more so among the *M. tuberculosis* strains, the significant variation existing among the phenotypes, can be attributed to their transcriptome regulation [13,14]. Alvarez-Eraso et al. [15] revealed that clinical strains of *M. tuberculosis* induced distinct growth phenotype when Mcr11 sRNAs were silenced in two Colombian clinical strains (UT127 and UT205) and a laboratory H37Rv strain. This study further revealed that the H37Rv and UT127 strains showed a delayed growth phenotype, while growth was inhibited in the UT205 strain, which also exhibited the clumping phenotype. Recently, Jumat and Chin [16] revealed the presence and differential expression of novel sRNAs in MTBC lineage 1 strains compared to other lineages. Understanding the underlying molecular mechanisms controlling the MTBC strains, particularly those involved in virulence and pathogenicity, would contribute to the development of novel strategies that can be applied to the treatment of TB disease [17]. To date, there is limited information and understanding of the MTBC non-coding RNAs, and even fewer have been functionally characterized, with only 2 studies [15,16] that investigated strains of clinical importance. Due to the significant role of non-coding RNAs (ncRNAs) in the modulation of the pathogen’s transcriptome, these transcripts are an ideal target to explore for novel therapeutic targets as they may greatly contribute to the lineage-specific transcriptome regulation. The current study used next-generation transcriptomics tools to investigate the presence and differential regulation of coding and ncRNAs in clinical strains of F15/LAM4/KZN and Beijing strains genotype families of the Euro-American and East-Asian lineages that are dominant within the South African population, respectively. Furthermore, strain-specific gene regulatory mechanisms and other novel ncRNAs that are critical in transcriptome regulation of these clinical strains of *Mycobacterium tuberculosis* were elucidated.

## 2. Results and Discussion

Despite high genetic similarities among the clinically relevant *M. tuberculosis* strains, their virulence traits and pathogenicity [18,19], in vitro and in vivo host responses [20,21,22,23,24], and other pathogen-specific phenotypic differences [25,26,27] indicate complex molecular mechanisms that govern their gene regulation. Previous studies revealed novel noncoding RNAs in clinically relevant strains of *M. tuberculosis* [15,16]. Hence, the current study investigated differentially expressed transcripts to interrogate the presence and abundance of noncoding RNAs in the globally present Beijing strain and the KZN strain predominantly represented in the KwaZulu-Natal region of South Africa compared to the laboratory H37Rv strain.

### 2.1. The Beijing Strain Induced a Higher Number of Significantly Differentially Expressed Transcripts Compared to the F15/LAM4/KZN Strain

*DeSeq2* analysis revealed a total of 856 and 1095 significantly differentially expressed transcripts (SDETs) between the KZN and Beijing strains, compared to the laboratory H37Rv, respectively, from the Total RNA Seq reads. The Beijing vs. H37Rv strains exhibited the highest number of both up (630) and downregulated (465) transcripts compared to KZN vs. H37Rv up (518) and downregulated (338) transcripts in Total RNA Seq reads. This was comparable to the high SDETs in smRNA Seq outputs with a total of 423 and 524 transcripts in KZN vs. H37Rv and Beijing vs. H37Rv, respectively, for both annotated and unannotated transcripts. *DeSeq2* analysis revealed the presence of well-annotated transcripts in these clinical strains as well as unknown transcripts that varied among the clinical strains compared to the H37Rv strain (Table 1). Furthermore, the SDETs were intersected through a Venn diagram (Figure 1A), and 428 transcripts were common to both strains vs. H37Rv, while 428 and 667 transcripts were exclusively differentially regulated by the KZN vs. H37Rv and Beijing vs. H37Rv, respectively. The up- and down-regulated SDETs among the strains enriched various KEGG pathways as shown in Figure 1B–F. The peptidoglycan biosynthesis KEGG pathway was enriched in the common transcripts as well as in the upregulated transcripts for both the Beijing and KZN strains, versus the laboratory, H37Rv strain. Furthermore, the upregulated transcripts for both strains enriched the Microbial metabolism in diverse environments and lysine degradation KEGG pathways. Other KEGG pathways for the down-regulated transcripts were unique as shown in Figure 1D,F.

The observation of a greater number of SDETs in the Beijing strain compared to the laboratory H37Rv, might suggest a more complex and adaptive transcriptional responses in this strain compared to the KZN strain. Previous studies revealed a hypervirulent phenotype of the Beijing strain on in vitro [28] and in vivo infection models [29,30], which is characterized by low cytokine responses [24,31]. Additionally, its global prevalence is associated with high transmissibility [32,33] compared to other clinical strains. Hence, transcriptional plasticity rendered by high SDETs identified in the current study might be advantageous to this clinical strain. The current study also revealed that the Beijing strain uniquely upregulated transcripts involved in the Tuberculosis KEGG pathway, which include pathogen- and host-specific transcripts that play critical roles in pathogenesis, while the KZN strain upregulated transcripts were involved in fatty acid and steroid degradation KEGG pathways. The presence of uniquely enriched KEGG pathways in the KZN strain suggests that despite its lower number of SDETss compared to the Beijing strain, it may possess distinct mechanisms that contribute to its persistence and virulence in specific populations, and this remains to be investigated in genetically diverse strains and infected individuals. The uniqueness of the KEGG pathways induced by these clinical strains could reflect strain-specific adaptations that confer advantages in particular environments or host interactions. It has been established that these strains induce unique immune responses [24,27,34,35]. Hence, their ability to activate unique transcriptional pathways is necessary for their specific phenotype and behavior. Furthermore, it was not surprising that common transcripts and those unique for each strain all enriched the peptidoglycan biosynthesis pathway. This pathway is essential for structural support, cellular processes, and virulence; hence, its enrichment is crucial in all *M. tuberculosis* strains [36,37].

### 2.2. Clinical Strains of M. tuberculosis Exhibit High Antisense Regulatory ncRNA Transcripts

The SDETs identified by *DeSeq2* identified ~50% of transcripts that were differentially expressed in unannotated regions and, hence, were termed “unknown”. Thus, *Rockhopper,* which has been widely used specifically for bacterial RNA Seq data to characterize novel RNA transcripts [38], especially for the smRNA Seq data, was used for this analysis. *Rockhopper* revealed the presence of a varying number of cis (antisense)- and trans (sense)-encoded ncRNAs in both smRNA Seq and Total Seq data, as depicted in Table 2. A high number of the antisense noncoding RNAs was observed, compared to the sense RNAs in both smRNA Seq and total RNA Seq reads, and these varied among the strains. The KZN strain had the highest number of the sense (2310) and antisense (9590) RNAs from the smRNA Seq reads in contrast to the Beijing strain that exhibited the highest number of both sense (29) and antisense (370) ncRNAs in all Total RNA Seq reads, both compared to the laboratory strain, respectively (Table 2; Appendix A.

Despite the high number of antisense RNA predicted by *Rockhopper*, it should be noted that many mRNA target transcripts had multiple regulatory antisense ncRNAs. Lloréns-Rico et al. [39] stated that many of these ncRNA expressions may not induce significant changes in gene expression because their expression may be due to transcriptional noise arising at spurious promoters. Thus, only ncRNA transcripts with high RPKM may have biological significance. Thus, the target mRNA transcripts for the top 5 expressed antisense RNAs were investigated in both smRNA and total RNA Seq data (Appendix A). All strains exhibited high expression of the large (>1300 bases) 5S, 16S, and 23S rRNA antisense ncRNA in Total Seq data. The smRNA Seq top 5 antisense transcripts targets were as follows: H37Rv (HAD hydrolase, family IB, OmpR family two-component system sensor histidine kinase, OmpR family two-component system response regulator, integral membrane protein, drug:H+ antiporter-2 (14 Spanner) (DHA2) family drug resistance MFS transporter); Beijing (two-component system response sensor kinase, hypothetical protein, integral membrane protein, transmembrane protein, hypothetical protein, PemK-like protein); KZN (two-component system sensor kinase phoR, hypothetical protein, PemK protein, phiRv2 phage protein X 4, fatty-acid-CoA ligase FadD22). Furthermore, the smRNA Seq transcripts detected by *Rockhopper* included annotated protein-coding transcripts, tRNA, novel small RNAs, as well as previously identified sRNAs such as *ASdes*, *ASpks*, and *AS1726* that are antisense to *desA1*, *pks12*, and *Rv1726* genes, respectively.

Despite many studies exploring the transcriptional landscape of clinical strains of *M. tuberculosis* [40,41,42,43], the presence and the role of regulatory ncRNAs remained unexplored, especially since the large number of transcripts map to unknown regions as detected by *DeSeq2* in the current study. *Rockhopper’s* ability to detect and characterize both known and unknown transcripts from bacterial RNA-Seq data [44] was crucial in shedding light on this previously unexplored dimension of the *M. tuberculosis* transcriptome. One of the notable findings from the Total RNA Seq data was the high expression of large (>1300 bases) antisense ncRNAs targeting rRNAs (5S, 16S, and 23S rRNA transcripts). Ribosomal RNA is central to the protein synthesis machinery [45], and antisense ncRNAs targeting rRNAs could potentially interfere with ribosome assembly or function, thereby affecting global protein synthesis. This suggests that antisense ncRNAs might have a broad impact on cellular processes by modulating rRNA activity. The biological relevance of these antisense RNAs in regulation of translational machinery under stress conditions, warrants further investigation in clinical strains of *M. tuberculosis* and their potential exploitation in anti-TB therapies [46].

A significant finding in the present study is that the number of antisense ncRNAs was more than double the number of sense ncRNAs, which was comparable to previous studies [16,47,48]. These antisense ncRNAs targeted well-annotated protein-coding genes as well as hypothetical proteins. For instance, the smRNA Seq data revealed that the top 5 antisense ncRNAs targeted genes involved in critical functions, such as drug resistance in H37Rv [49], signal transduction in Beijing and KZN strains, and fatty acid metabolism in the KZN strain [50]. Recently, the KZN strain was shown to downregulate the *fas* transcript that converts malonyl-CoA to fatty acids [27] during growth in cholesterol-rich media; hence, significant upregulation of fatty acids antisense ncRNAs may be involved in this regulation. These findings suggest that antisense ncRNAs may play a regulatory role in fine-tuning important cellular processes, particularly under stress conditions such as antibiotic exposure or host lipids. The KZN strain exhibited the highest number of sense (2310) and antisense (9590) ncRNAs in the smRNA Seq data, indicating particularly complex regulatory networks by exploiting these ncRNAs for this strain. This extensive ncRNA repertoire could be linked to the KZN strain’s ability to adapt to various stress conditions, such as invasion and replication of in vitro infection models [34,51] or its potential for persistence within the host as one of the dominant strains in the KwaZulu-Natal province of South Africa during the Tugela ferry outbreak [3,52]. The Beijing strain showed the highest numbers of both sense (29) and antisense (370) ncRNAs in the Total RNA Seq data. This suggests that while the Beijing strain may not express as many ncRNAs as the KZN strain, it may still express a robust set of regulatory ncRNAs that are crucial for its survival and pathogenicity as a globally prevalent lineage [53,54].

### 2.3. Highly Expressed Sense ncRNAs Regulate Metabolism and Cell Wall Processes in Clinical Strains of M. tuberculosis

The sense ncRNAs (trans-encoded) are transcribed distantly from their target genes and tend to have multiple mRNA targets due to their partial base-pairing complementarity [55]. In the present study, fewer sense-encoded ncRNAs compared to antisense ncRNA were observed for all strains (Table 2, Appendix A). The top 10 most expressed sense ncRNAs are shown in Table 3, Table 5, and Table 7 for the KZN, Beijing, and H37Rv strains, respectively. All smRNA Seq transcripts shown in Table 4, Table 6 and Table 8 have been named based on the transcript length (e.g., Transcript 161 in Table 4). The three most expressed ncRNAs from smRNA Seq and the one highly expressed ncRNA from Total Seq were characterized by elucidating their predicted secondary structures, target mRNAs as well as target mRNA functions. The KZN strain Transcript 161 had a total of 139 mRNA targets with the highest probability of binding to *Rv1000c*, which is a conserved hypothetical protein in *M. tuberculosis*. The 139 mRNA targets had 58 known and predicted protein-protein interactions with each other (Figure 2A) and were functionally categorized to be involved in Mycobacterial pentapeptide repeat and lipid transport proteins (Figure 2B). Transcript 85 had no significant mRNA target while Transcript 64 and Transcript 433 targeted *Rv2847c* and *Rv1296*, respectively, which are both involved in intermediary metabolism and respiration (Table 4). The Beijing strain most expressed Transcript 354 had 130 significant mRNA targets with the highest probability of binding to *Rv1000c*. The 130 transcript 354 targets largely formed two protein-protein network clusters with 33 (Figure 3A) and 32 genes (Figure 3B) that enriched several functional categories including mixed Mycobacterial pentapeptide repeats and PE family, Rhodanese homology domain as well as the HNH nuclease (Figure 3C). Transcript 208, transcript 297, and transcript 436 had the highest probability of binding to *Rv3709c*, *Rv1031,* and *Rv2155c*, respectively, which are mostly involved in intermediary metabolism and respiration, as well as cell wall and cell processes (Table 6). It is intriguing to observe the contrast between the clinical strains’ most expressed sense ncRNAs compared to the laboratory strain. The two highest sense ncRNAs (Transcript 124 and Transcript 133) revealed no significant mRNA targets while Transcript 156 and Transcript 470 targeted *Rv0061c* (hypothetical protein) and *Rv0124* (PE_PGRS2) (Table 8). The RPKMs of the target mRNAs for the highly expressed sense ncRNAs for all strains were low, ranging from 10 to 281, suggesting a regulatory mechanism of these mRNAs at a transcription level (Table 4, Table 6, and Table 8).

**Table 3 ijms-26-00217-t003:** Top 10 most expressed noncoding RNAs identified in the KZN strain and their respective transcript location and predicted length.

Transcription Start	Transcription Stop	Strand	Expression (RPKM)	Transcript Length
4087068	4086907	−	119182	161
4340712	4340797	+	8378	85
2335625	2335561	−	5538	64
3124881	3124715	−	5014	166
613142	613226	+	3280	84
65640	65532	−	3015	108
2366409	2366548	+	2846	139
1432800	1432621	−	2832	179
2744204	2744067	−	2540	137
3584108	3584007	−	2409	101

**Table 4 ijms-26-00217-t004:** Functional characterization of the most expressed sRNAs in the KZN strain and their respective mRNA targets and function.

	Target mRNA/s	Target mRNA Expression and Function	*p*-Value	Probability	Energy (kcal/mol)
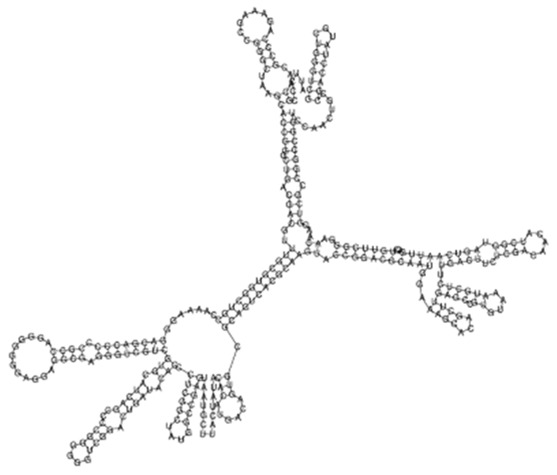 smRNA Seq Transcript 161 −123.10 kcal/mol	*Rv1000c* Total = 139	RPKM = 39 Conserved hypothetical protein	0.000504	0.797	−43.47
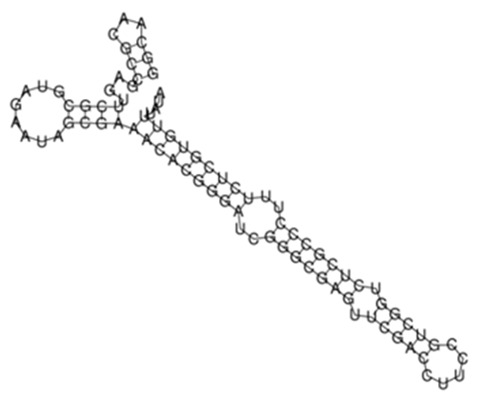 smRNA Seq Transcript 85 −42.68 kcal/mol	No mRNA targets	-	-	-	-
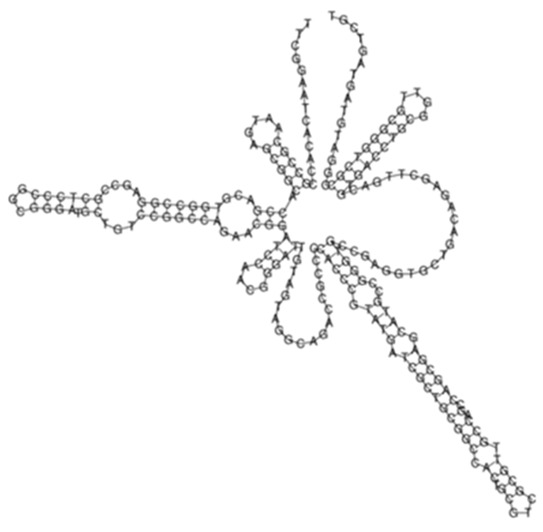 smRNA Seq Transcript 64 −104.24 kcal/mol	cysG (*Rv2847c*) Total = 4	RPKM = 28 Intermediary metabolism and respiration	2.7 × e^−04^	0.67	−46.38
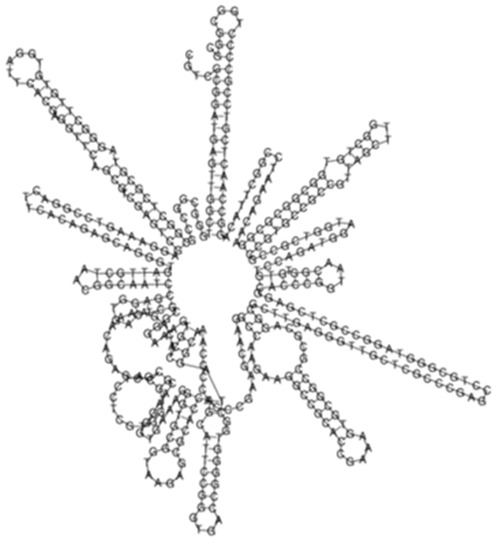 Total Seq Transcript 433_7H9 −224.81 kcal/mol	thrB (*Rv1296*) Total = 165	RPKM = 208 Intermediary metabolism and respiration	3.8 × e^−04^	0.85	−57.15 kcal/mol

**Table 5 ijms-26-00217-t005:** Top 10 most expressed noncoding RNAs identified in the Beijing strain and their respective transcript location and predicted length.

Transcription Start	Transcription Stop	Strand	Expression Beijing (RPKM)	Transcript Length
4099112	4098758	−	55,889	354
4153112	4153320	+	8389	208
1286909	1287206	+	6450	297
1166794	1166689	−	4574	105
333401	333325	−	3969	76
2445202	2445159	−	3390	43
1728439	1728505	+	3310	66
2546571	2546665	+	3284	94
2052499	2052601	+	3276	102
3362906	3362670	−	2961	236

**Table 6 ijms-26-00217-t006:** Functional characterization of the most expressed sRNAs in the Beijing strain and their respective mRNA targets and function.

	Target mRNA/s	Target mRNA Expression and Function	*p*-Value	Probability	Energy (kcal/mol)
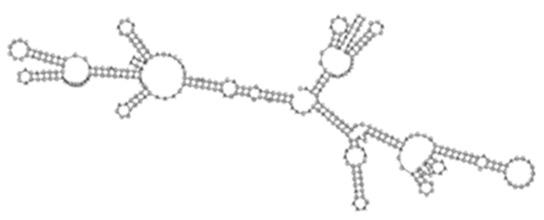 smRNA Seq Transcript 354 −153.60 kcal/mol	*Rv1000c* Total = 130	RPKM = 84 Conserved hypothetical protein	4.9 × e^−04^	0.80	−43.04
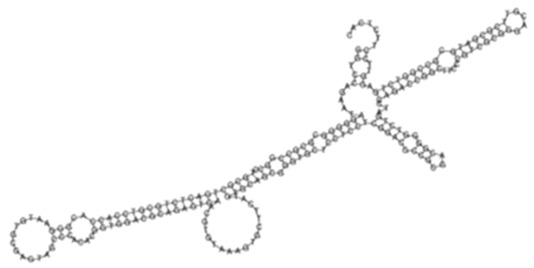 smRNA Seq Transcript 208 −106.80 kcal/mol	ask (*Rv3709c*) Total = 59	RPKM = 79 Intermediary metabolism and respiration	2.5 × e^−03^	0.92	−280.09
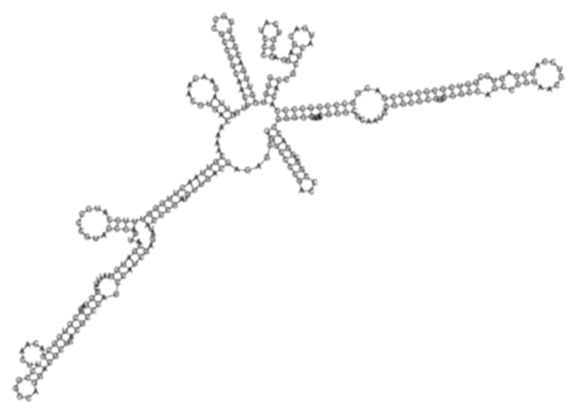 smRNA Seq Transcript 297 −118.90 kcal/mol	kdpC (*Rv1031*) Total = 3	RPKM = 91 Cell wall and cell processes	2.6 × e^−04^	0.60	−43.43 kcal/mol
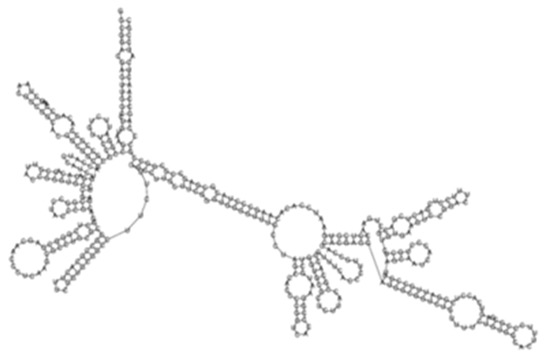 Total Seq Transcript 436_7H9 −184.80 kcal/mol	murD (*Rv2155c*) Total = 159	RPKM = 19 Cell wall and cell processes	9.6 × e^−04^	0.82	−42.11

**Table 7 ijms-26-00217-t007:** Top 10 most expressed noncoding RNAs identified in the laboratory H37Rv strain and their respective transcript location and predicted length.

Transcription Start	Transcription Stop	Strand	Expression (RPKM)	Transcript Length
3907698	3907574	−	15,406	124
612907	613040	+	7520	133
65536	65380	−	5360	156
1923729	1923835	+	4942	106
3363210	3363041	−	3932	169
4093743	4093810	+	3585	67
3726140	3726280	+	3547	140
2056349	2056213	−	3326	136
2150827	2150904	+	3268	77
3753558	3753668	+	3151	110

**Table 8 ijms-26-00217-t008:** Functional characterization of the most expressed sRNAs in the laboratory H37Rv strain and their respective mRNA targets and function.

	Target mRNA/s	Target mRNA Expression and Function	*p*-Value	Probability	Energy (kcal/mol)
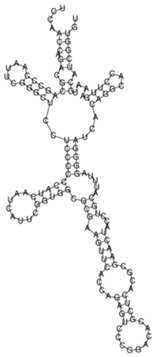 smRNA Seq Transcript 124 −34.20 kcal/mol	No mRNA targets	-	-	-	-
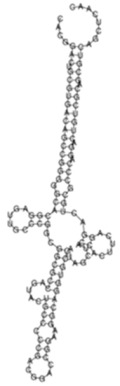 smRNA Seq Transcript 133 −−52.70 kcal/mol	No mRNA targets	-	-	-	-
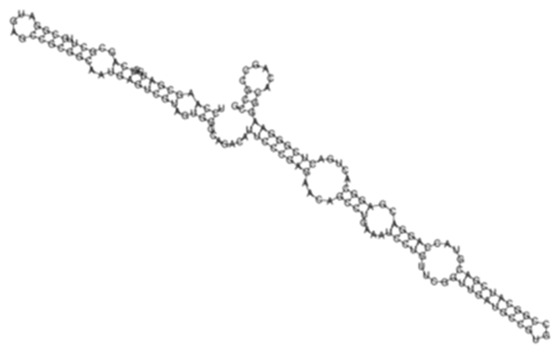 smRNA Seq Transcript 156 −65.60 kcal/mol	*Rv0061c* Total = 1	RPKM = 281 Hypothetical protein	1.1 × e^−09^	0.94	−227.44
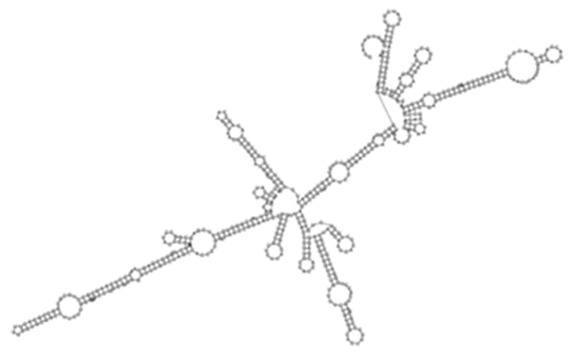 Total Seq Transcript 470_7H9 −244.20 kcal/mol	PE_PGRS2 (*Rv0124*) Total = 159	RPKM = 10 PE/PPE	1.3 × e^−04^	0.92	−41.64

Despite progress made in genomics, bioinformatics, and other analytical tools, *M. tuberculosis* still possesses >800 hypothetical proteins that remain to be functionally characterized in both laboratory and clinically relevant strains [56,57]. The KZN and Beijing strains’ most expressed sense ncRNAs targeted the *Rv1000c* gene, which is a conserved hypothetical protein with unknown function in *M. tuberculosis*. Upon analysis of the *Rv1000c* operon, only one characterized gene was identified as a Probable arginine deiminase ArcA (arginine dihydrolase) known as *Rv1001*, while the rest of the genes (*Rv0999*, *Rv0998*, *Rv0997*, *Rv1002c*) encoded hypothetical proteins. Sürken et al. [58] revealed that the deletion of *Rv1001* did not affect arginine metabolism during aerobic conditions and in chronic infection of mice models. It can be extrapolated that transcripts in this operon are involved in metabolism and the expression of sense ncRNAs in both clinical strains can be a molecular mechanism that regulates specific metabolic processes that are provided by this operon. The *Rv2847c*, *Rv1296,* and *Rv3709c* transcripts that were targets of highly regulated ncRNAs expressed by the KZN and Beijing strains, were elucidated to be involved in intermediary metabolism and respiration category. This is similar to the recent findings reported by Jumat and Chin [16] in Lineage 1 of *M. tuberculosis* novel sRNA_1 and sRNA_29 that targeted genes involved in metabolism and respiration. It is thus clear, that *M. tuberculosis* intracellular metabolic processes are regulated by ncRNAs and remain a potential target of exploration for anti-TB therapies [59].

Previous studies [26,60,61,62] have shown that Beijing strains possess unique cell wall lipids that are involved in provoking host immune responses. Hence, in the current study, it was not surprising to identify highly expressed ncRNAs that are involved in cell wall and cell processes for this strain as these regulatory RNAs could be involved in lipid differentiation in the Beijing strain. The KZN strain Transcript 161 and the Beijing strain Transcript 345 mRNA targets formed protein-protein networks that were functionally categorized into mycobacterial pentapeptide repeats, PE family, Rhodanese homology domain as well as the HNH nuclease. The high enrichment of mycobacterial pentapeptide repeats by the KZN and Beijing strain mRNA targets for both highly expressed ncRNAs could be associated with exploiting these transcripts to contribute to antibiotic resistance. Previous studies [63,64] have shown that *M. tuberculosis* pentapeptide, MfpA is associated with the development of drug resistance through inhibition of DNA gyrase activity during fluoroquinolone treatment. Hence, the ncRNA transcripts identified in the current study may serve as ideal targets for novel antimycobacterial therapies. The lack of significant mRNA targets for the H37Rv strain does not indicate biological insignificance since ncRNA structural features may still contribute to their role in regulatory networks by acting as sponges for other regulatory RNAs or proteins [65]. It was also interesting to observe that H37Rv-specific ncRNAs target hypothetical protein and PE_PGRS2 of the PE/PPE functional categories that are known for their critical role in host–pathogen interaction [66], hence, this ncRNA transcript may have been involved in low induction of MCP-1, IL17, and TNF-α during infection of pulmonary epithelial cells at 72 hrs post-infection by the H37Rv strain compared to the KZN and Beijing strains [24] and this mechanism remains to be elucidated in future gene knockout studies.

### 2.4. The Laboratory Strain Exhibits High Expression of the Common Sense ncRNAs Expression

Evaluation of the differential expression of sense-encoded RNAs elucidated 392 significantly (*q* < 0.05) differentially expressed sense ncRNAs between the KZN vs. H37Rv strain, compared to 337 in the Beijing vs. H37Rv (Figure 4A, Appendix A). Among the differentially expressed transcripts, 36 common sense ncRNAs were observed among the strains, with diverse expression profiles (Figure 4B, Appendix A). Based on the hierarchical clustering of the common ncRNAs, the H37Rv induced the highest RPKMs of sense ncRNAs common transcripts while the KZN strain clustered separately with the lowest expression of most transcripts compared to H37Rv and the Beijing strains (Figure 4B). The highest expression difference among the strains was observed in transcript_130 (Figure 4C) with 0, 1770, and 6092 RPKMs for KZN, Beijing, and H37Rv strains, respectively (Appendix A). Transcript_130 exhibited a high probability of interaction with the *Rv2225* gene with thermodynamic energy of interaction −26.17 kcal/mol (Figure 4D).

It is intriguing that the differential transcriptome patterns are not only observed in in vitro [67,68] and in vivo [69] infection models during exposure to clinically relevant MTBC strains but can be traced to regulatory ncRNAs as shown in the current study for the commonly shared transcripts. The most notable difference was the expression of Transcript_130 that targets the *Rv2225* gene encoding the 3-methyl-2-oxobutanoate hydroxymethyltransferase PanB. This enzyme is involved in transforming α-ketoisovalerate (3-methyl-2-oxobutanoate) to ketopantoate (2-dehydropantoate) through the transfer of a hydroxymethyl group provided by N 5,N 10-methylenetetrahydrofolate cofactor in the presence of a metal ion [70,71]. This finding suggests that the H37Rv strain may exploit Transcript_130 in regulating this metabolic process compared to the KZN and Beijing strains. The strain-specific differences in ncRNA expression, particularly between clinical strains and the laboratory strain H37Rv, emphasize the importance of studying clinical isolates to gain a better understanding of *M. tuberculosis* pathogenesis. These differences also highlight the potential limitations of relying solely on laboratory strains for studying tuberculosis biology and developing therapeutics. Future studies will exploit molecular tools and laboratory-based validation studies including loss-of-function and gain-of-function experiments to functionally characterize the identified ncRNAs in the current study and their respective roles in *M. tuberculosis* physiology to provide supporting evidence to propose these transcripts as potential targets for anti-TB exploration studies. Furthermore, the protein products of the targeted mRNAs will be profiled to confirm the regulatory roles of the highly expressed ncRNAs that were identified in the current study.

## 3. Materials and Methods

### 3.1. Mycobacterium tuberculosis Strains and Culture Conditions

The F15/LAM4/KZN and Beijing strains of the Euro-American and East-Asian lineages that were previously isolated from patients [3,52,72] were grown in supplemented Middlebrook 7H9 media (Becton Dickinson, Durban, South Africa) and their genotype families established by means of IS6110-restriction fragment length polymorphism analysis (RFLP) and spoligotyping [73]. The laboratory strain *M. tuberculosis* H37Rv (ATCC 25618) was obtained from the University of KwaZulu-Natal (Medical Microbiology) culture collection and included in this study as a virulent control. All *M. tuberculosis* strains were grown in Middlebrook 7H9 broth (Becton Dickinson, Durban, South Africa) supplemented with 50% glycerol, 20% Tween-80 (Sigma-Aldrich, Durban, South Africa), and 10% OADC (Becton Dickinson, Durban, South Africa), and incubated in a shaking incubator at 37 °C until the logarithmic growth phase was reached [74].

### 3.2. Total RNA Isolation

Bacterial cultures were collected during the logarithmic growth phase (OD_600nm_: 0.8–1) and centrifuged (Labotec, Durban, South Africa) for 1 min and 30 s at 10,000× g. The supernatant was discarded, and 1 mL of TRI reagent (Zymo Research, Inqaba Biotec, Gauteng, South Africa) was followed by RNA extraction using Zymogen DirectZol RNA Miniprep kit Zymo Research, USA that enriched for both sRNAs and mRNA as detailed by Mvubu et al. [75]. Briefly, *M. tuberculosis* cultures suspended in TRI reagent containing 0.1 mm diameter zirconia beads (BioSpec, Bartlesville, OK, USA) were lysed with a high-speed bead beater (Precellys 24, Durban, South Africa). The lysate was centrifuged, and the RNA in the supernatant was precipitated with ice-cold ethanol (Sigma-Aldrich, Durban, South Africa) and cleaned using the RNA wash buffer within the DirectZol column. The RNA was on-column DNAse treated, followed by 2× wash steps to remove residual contaminants. The RNA was eluted with 30 µL of DNase/RNase-free water and quantified using the Nanodrop 2000c Spectrophotometer (Thermo Scientific, Gauteng, South Africa). The RNA quality and integrity were evaluated by 3-(N-morpholino) propane-sulfonic acid (MOPS) (Sigma-Aldrich, Durban, South Africa) gel electrophoresis [76] and Bioanalzyer (Admera Health, South Plainfield, NJ, USA). The RNA isolated from the laboratory H37Rv reference strain was used as a control, and all RNA samples were stored at −80 °C until sequencing.

### 3.3. Library Preparation and RNA Sequencing

The RNA-Seq library for the 3 biological replicates of cultures grown in Middlebrook 7H9 (Becton Dickinson, Durban, South Africa) media was prepared for the Illumina HiSeq 2000 sequencing platform (Illumina, Gauteng, South Africa) using the NEBNext Ultra II with RiboZero Plus kit (New England Biolabs, Gauteng, South Africa) with a sequencing depth of 60 million total pair-end reads per sample for Total RNA Seq. The TruSeq smRNA (Illumina, Gauteng, South Africa) kit was used for strains cultured in 7H9 for sequencing with a depth of 20 M pair-end reads per sample following the manufacturer’s instructions. In summary, 200 ng of total RNA in 50 µL was mixed with equal volumes of RNA purification magnetic beads. The resulting fragmented RNA was converted to double-stranded cDNA that was repaired, tailed, and ligated with indexed adaptors. The subsequent adaptor-linked cDNA library was amplified by PCR, purified and electrophoresis was performed on an Agilent high-sensitivity DNA chip for quality control. Each sample lane for the 100 bp sequencing (mRNA) was pooled in equimolar concentration and sequenced for pair-end reads of 100 cycles. All sequencing experiments were performed at the Admera Health sequencing facility (USA).

### 3.4. Read Mapping, Transcript Assembly, and Differential Expression

The quality of the reads was assessed by FastQC (v. 0.11.9) [77], and reads were trimmed and processed using Trimmomatic (v. 0.39) [78] to remove adapters and low-quality bases. The HiSat2 (v. 2.1.1.) software was used to map sequence reads against the *M. tuberculosis* H37Rv reference genome [79] within KBase Bioinformatics analysis interphase [80]. StringTie (v. 2.1.4) was used to assemble and quantify the transcripts [81,82]. DeSeq2 (v. 1.45.0) was used to calculate differential expression and fold changes among the KZN and Beijing clinical strains compared to the laboratory H37Rv strain using the false discovery rate (FDR) *q* < 0.05 [83]. The Nextera adapter option was turned on for Trimmomatic, while default parameters were used for HiSat2, StringTie, and DeSeq2 bioinformatics tools. The raw RNA Seq reads are openly available in Sequence Read Archive, reference number SUB10671416.

### 3.5. Non-Coding RNA Detection

Rockhopper (v2.03) used reference-based analysis from well-annotated strains of *M. tuberculosis* to predict ncRNAs (sRNA and long non-coding RNAs), together with their genomic locations as well as their respective operons from trimmed RNA Seq fastq files [44]. *M. tuberculosis* H37Rv (CP003248.2), *M. tuberculosis* str. Beijing/NITR203 (478739819) and *M. tuberculosis* KZN605 (392051818) sequences downloaded by Rockhopper from RefSeq using their accession numbers were used as background references for each strain. Rockhopper used strand-guided parameters to reveal the genomic location and strand orientation of the identified noncoding RNAs. Genomic locations of the detected noncoding RNAs were visualized using Integrative Genomics Viewer (IGV) (v2.17.4) [84], and noncoding RNA sequences were extracted for further characterization. Lastly, transcript abundance was elucidated and displayed as Reads Per Kilobase per Million mapped reads (RPKM) that is normalized by the upper quartile of gene expression instead of the total number of reads for the selected transcript.

### 3.6. In Silico Characterization of Regulatory Noncoding RNA Transcripts

The strain-specific highly expressed ncRNAs sequences were extracted using genomic coordinates and read abundance spanning the predicted novel ncRNAs in IGV and in silico characterized using default settings in RNAfold web server http://rna.tbi.univie.ac.at//cgi-bin/RNAWebSuite/RNAfold.cgi (accessed on the 2nd of November 2024) [85]. Their respective mRNA targets were identified by *TargetRNA3* [86], and the transcript with the highest probability of interaction and lowest *p*-value was selected for further visualization. The target mRNA interaction network and graphic outputs were generated using the Search Tool for Retrieval of Interacting Genes/Proteins (STRING) database [87]. Venny [88] and Multi Experiment Viewer (MeV) [89] were used for gene expression data visualization for differential expression of noncoding RNAs between the *M. tuberculosis* strains.

## 4. Conclusions

In conclusion, this study provides new insights into the coding and ncRNA landscape of clinically relevant *M. tuberculosis* strains through transcriptomics and bioinformatics tools. The repertoire of ncRNAs and their targets indicate more complex molecular mechanisms that have been significantly underestimated in *M. tuberculosis*, especially for strains that are highly transmissible and dominant globally. Furthermore, coding and non-coding transcripts stimulated relevant virulence-associated pathways such as peptidoglycan synthesis and cellular functions, respiration, metabolism, ribosome function and drug resistance, highlight the importance of both common and strain-specific pathways in *M. tuberculosis*. Future studies should focus on characterizing the functional roles of the identified ncRNAs, to better understand their contributions to *M. tuberculosis* survival and pathogenicity in different infection models. Furthermore, exploring the potential of targeting ncRNAs for therapeutic purposes could open new avenues for the treatment of tuberculosis, particularly for drug-resistant strains.

## Figures and Tables

**Figure 1 ijms-26-00217-f001:**
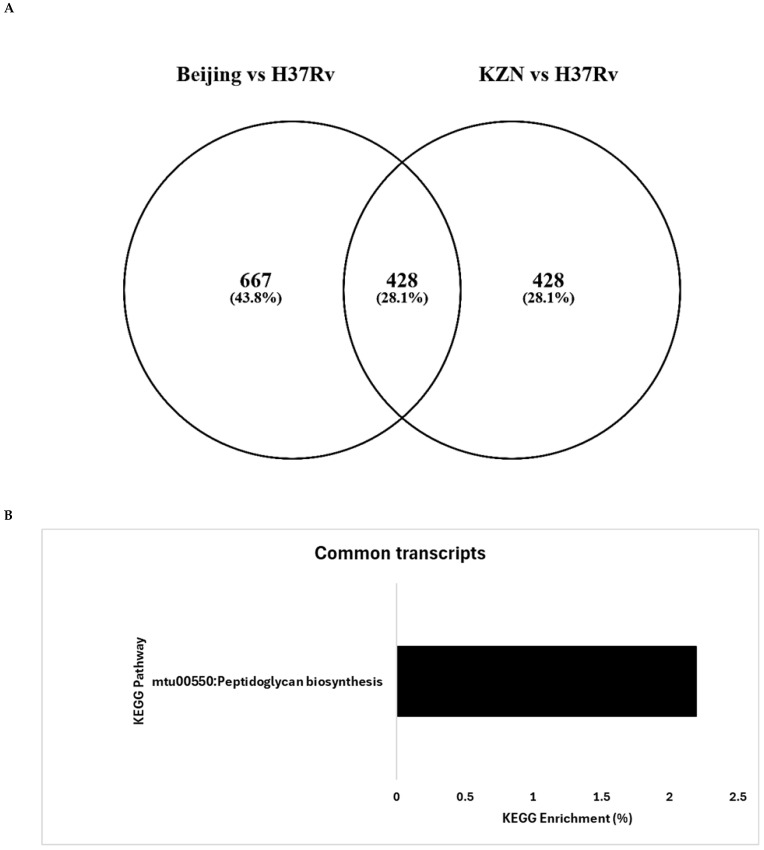
(**A**) A Venn diagram showing significantly (*q* < 0.05) differentially expressed transcripts in the KZN and Beijing strains when compared to the laboratory strain, H37Rv. Annotated transcripts that were differentially expressed were enriched within the Database for Annotation, Visualization, and Integrated Discovery (DAVID) and the 428 common transcripts were mostly involved in (**B**) peptidoglycan biosynthesis KEGG pathway. (**C**) Beijing vs. H37Rv-specific upregulated SDETs enriched KEGG pathways as follows, peptidoglycan biosynthesis, Tuberculosis, Lysine biosynthesis, and microbial metabolism in diverse environments, while (**D**) biosynthesis of siderophores group non-ribosomal peptides were enriched for the downregulated transcripts. (**E**) Peptidoglycan biosynthesis, Microbial metabolism in diverse environments, Steroid degradation, Lysine biosynthesis, and Fatty acid degradation, as well as the (**F**) Bacterial secretion system and Fructose and mannose metabolism KEGG pathways were enriched for KZN strain compared to the laboratory strain for up- and down-regulated transcripts, respectively. RNA was extracted from KZN, Beijing, and H37Rv strains grown in 7H9 and subjected to Total and smRNA Seq, followed by mapping the reads to the H37Rv strain using *HiSat2*, which were then assembled using *StringTie*. Differentially expressed transcripts were elucidated through *DeSeq2* tools, and the FDR-corrected *q* < 0.05 was used to select transcripts for KEGG pathway enrichment.

**Figure 2 ijms-26-00217-f002:**
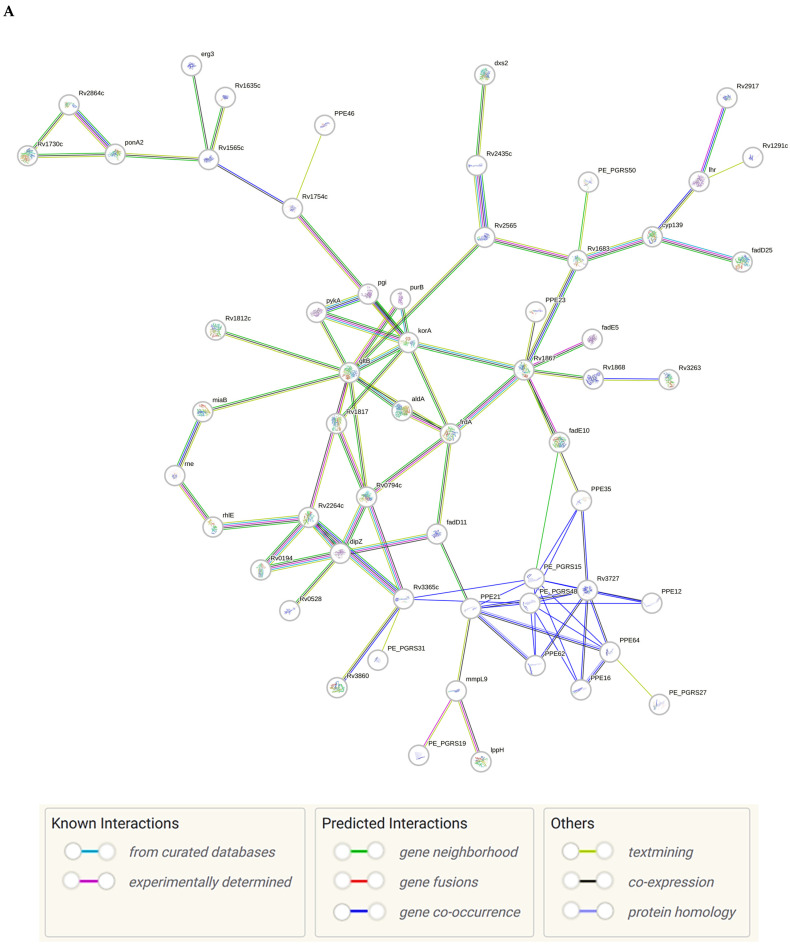
(**A**) Protein–protein network interactions for 58 out of 139 mRNA targets for Transcript 161. These interactions included known interactions, predicted interactions as well as others from the annotated *M. tuberculosis* genome. (**B**) Functional Enrichment analysis of the 58 interacting proteins revealed diverse functional categories including Mycobacterial pentapeptide repeat and lipid transport proteins. The 139 mRNA targets for Transcript 161 were used to generate the protein network and functional enrichment within the Search Tool for Retrieval of Interacting Genes/Proteins (STRING) database. FDR: False Discovery Rate.

**Figure 3 ijms-26-00217-f003:**
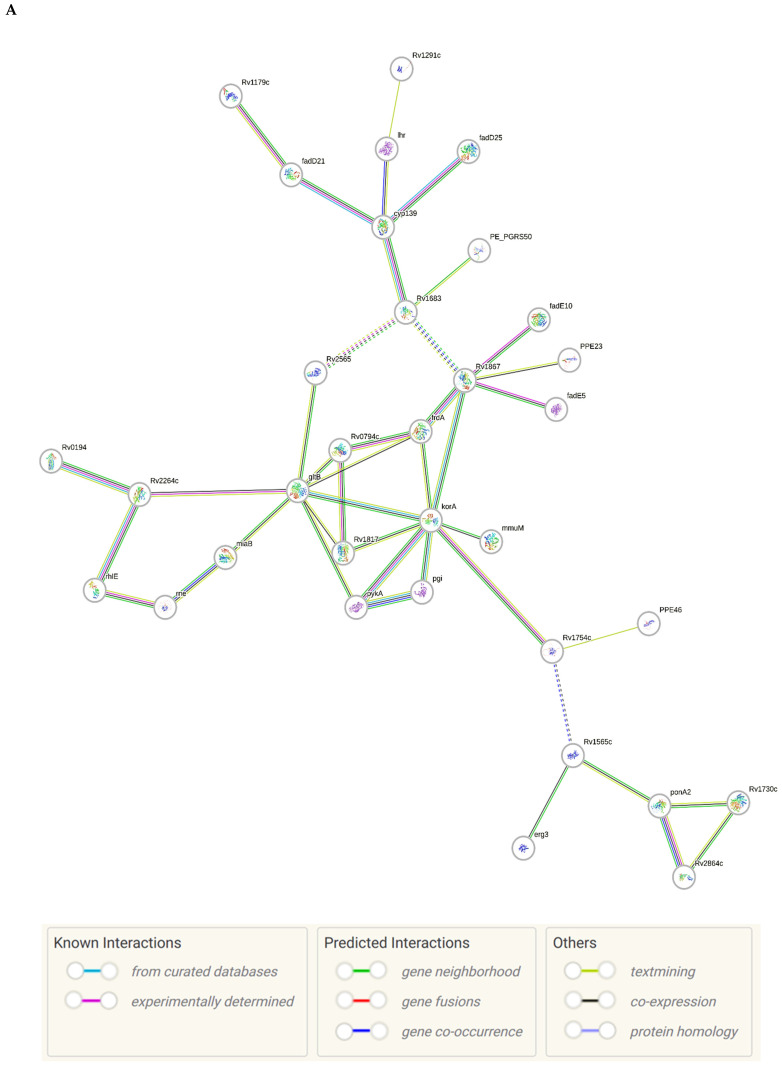
(**A**) Protein–protein network interactions for 33 out of 130 mRNA targets; and (**B**) 32 out of 130 mRNA targets for Transcript 354, respectively. These identified interactions included known interactions, predicted interactions as well as others from the annotated *M. tuberculosis genome*. (**C**) Functional Enrichment analysis of both protein networks revealed diverse functional categories, including mixed Mycobacterial pentapeptide repeats and PE family, Rhodanese homology domain, as well as the HNH nuclease. The 130 mRNA targets for Transcript 354 were used to generate the protein network and functional enrichment within the Search Tool for Retrieval of Interacting Genes/Proteins (STRING) database. FDR: False Discovery Rate.

**Figure 4 ijms-26-00217-f004:**
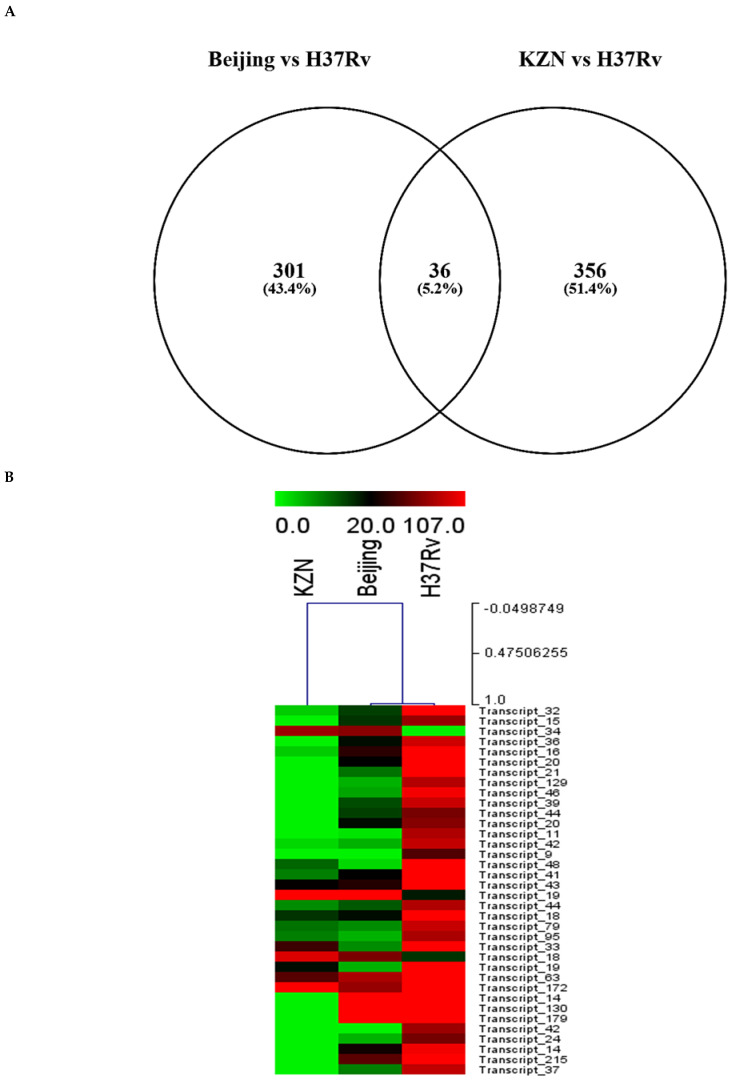
(**A**) A Venn diagram showing significantly (*q* < 0.05) differentially expressed sense ncRNAs in KZN vs. H37Rv and Beijing vs. H37Rv strains from *Rockhopper* Bioinformatics pipeline indicating shared and unique transcripts in each clinical strain compared to the laboratory strain. (**B**) A heatmap showing the expression patterns of 36 transcripts shared by all strains. The H37Rv strains exhibit higher expression of shared transcripts compared to other clinical strains and clustered closely to the Beijing strain. (**C**) The predicted ncRNA secondary structure of Transcript_130 with the predicted thermodynamic energy of −54.27 kcal/mol. (**D**) Transcript_130 exhibited a higher probability of interacting with the *Rv2225* gene with thermodynamic energy of interaction −26.17 kcal/mol.

**Table 1 ijms-26-00217-t001:** Significantly (*q* < 0.05) differentially expressed transcripts (SDETs) in two clinical strains of *Mycobacterium tuberculosis* versus the laboratory H37Rv strain.

	KZN vs. H37Rv	Beijing vs. H37Rv
Total RNA Seq All SDETs	856	1095
Upregulated known transcripts	257	308
Upregulated unknown transcripts	261	322
Downregulated known transcripts	158	219
Downregulated unknown transcripts	180	246
smRNA Seq All SDETs	423	524
Upregulated known transcripts	111	169
Upregulated unknown transcripts	110	143
Downregulated known transcripts	99	93
Downregulated unknown transcripts	103	119

**Table 2 ijms-26-00217-t002:** Total number of predicted RNA transcripts and untranslated regions by *Rockhopper* in smRNA and Total Seq reads of *M. tuberculosis* strains.

	KZN	Beijing	H37Rv
Total RNA Seq			
5′UTRs	47	38	516
3′UTRs	133	112	569
Sense predicted RNAs	10	29	13
Antisense predicted RNAs	195	370	244
Annotated protein coding transcripts	2426	2497	3178
Hypothetical proteins transcripts	1578	1616	861
rRNA	0	0	0
tRNA	45	45	45

smRNA Seq			
5′UTRs	926	888	893
3′UTRs	751	721	722
Sense predicted RNAs	2310	2154	2193
Antisense predicted RNAs	9590	4880	8241
Annotated protein coding transcripts	2426	2497	3178
Hypothetical proteins transcripts	1578	1616	861
rRNA	0	0	0
tRNA	45	45	45

## Data Availability

The raw RNA Seq reads used in this study are openly available in Sequence Read Archive, reference number SUB10671416.

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
