# Peer review of "Comparative Transcriptomics Reveal Differential Expression of Coding and Non-Coding RNAs in Clinical Strains of Mycobacterium tuberculosis"

_ijms, 2024, doi:10.3390/ijms26010217_

Round 1
Reviewer 1 Report
Comments and Suggestions for Authors
Mvubu et al. conducted an analysis of non-coding RNAs in three different Mycobacterium tuberculosis strains. Their study revealed that the strains exhibit distinct expression patterns in both genes and cis-/trans-acting non-coding RNAs. These patterns also highlighted strain-specific enrichment in distinct KEGG pathways. Furthermore, they suggested that non-coding RNAs may play a significant role in gene regulation.
In general, it is important and interesting to study non-coding RNA in gene regulation among different M. tuberculosis strains. However, this manuscript give a big picture of the molecular mechanisms of non-coding RNA, but detailed examples are missing. Although the manuscript included several different references to help demonstrate that non-coding RNA could regulate gene expression in M. tuberculosis, the authors did not provide specific evidence directly derived from this study. This made this manuscript look like a review instead of a research article. The authors could segregate the results and discussion into two sections.
Additionally, at 2.3 section, the authors uncovered that ncRNA regulated mRNA at transcription level. Do they further detect their expression level? Namely, protein level?
Below are some specific points.
Line 14: In the main text, the abbreviation for small RNA is written as “sRNA” (Line 38). Please ensure consistency throughout the manuscript.
Line 83: The authors stated that they used KZN strains in their analysis. Is "KZN" an abbreviation for F15/LAM4/KZN? Please clarify.
Lines 105–107: The sentence, “This was in contrast to low SDET…,” is unclear. Could the authors clarify what they are trying to demonstrate?
Line 105: Please define what is meant by "low SDET." Are there specific criteria? If "high SDET" is also used, please define it as well.
Table 1: The formatting is uneven. Please align the table properly.
Lines 112–118: This section is poorly written and difficult to follow. Please revise and polish the sentences for clarity.
Figure: The font sizes within the figure are inconsistent. Please ensure uniformity.
Lines 139–141: The content here was already mentioned in the previous paragraph. Repeating it is redundant and should be removed.
Comments on the Quality of English Language
Some sections are repetitive and overly wordy. Please revise to improve clarity and conciseness.
Author Response
College of Health Sciences
School of Laboratory Medicine and Medical Sciences
Medical Microbiology
Dr. Nicole Hansmeier
Editor-in-Chief
International Journal of Molecular Sciences
Dear Dr. Nicole Hansmeier
11 December 2024
Submission of manuscript Revision to International Journal of Molecular Sciences: Comparative transcriptomics reveal differential expression of coding and non-coding RNAs in clinical strains of Mycobacterium tuberculosis
We thank the Reviewers for their valuable comments. The queries and comments requested by reviewers are addressed in the point-by-point response below and highlighted in the manuscript file. Due to changes during manuscript edits, initial line numbers have changed.
Reviewers Comments
Reviewer 1
Reviewer comment
Mvubu et al. conducted an analysis of non-coding RNAs in three different Mycobacterium tuberculosis strains. Their study revealed that the strains exhibit distinct expression patterns in both genes and cis-/trans-acting non-coding RNAs. These patterns also highlighted strain-specific enrichment in distinct KEGG pathways. Furthermore, they suggested that non-coding RNAs may play a significant role in gene regulation.
In general, it is important and interesting to study non-coding RNA in gene regulation among different M. tuberculosis strains. However, this manuscript give a big picture of the molecular mechanisms of non-coding RNA, but detailed examples are missing. Although the manuscript included several different references to help demonstrate that non-coding RNA could regulate gene expression in M. tuberculosis, the authors did not provide specific evidence directly derived from this study. This made this manuscript look like a review instead of a research article. The authors could segregate the results and discussion into two sections.
Author Response
The manuscript has been revised following Reviewers suggestions and guidance. The revised sections have been highlighted for ease of identification in the manuscript document. It is unfortunate that there is very limited data in the literature that reports in this area of research. To date, only 2 studies (DOI: 10.1007/s11274-024-04089-6; 10.1007/s00284-021-02733-0) have used next generation sequencing to report on non-coding RNA in clinical strains of M. tuberculosis. Jumat and Chin (10.1007/s11274-024-04089-6) report on ncRNAs in Lineage 1 and Alvarez-Eraso et al (10.1007/s00284-021-02733-0) did not disclose the strain lineage for the identified sRNAs. Hence, the data generated in the current study could not be compared to many previous studies. Due to this limitation, the current study investigated coding and non-coding ncRNAs in 2 very dominant strains in KwaZulu-Natal of Lineage 2 and Lineage 4 (South Africa) and reported the genomic location of these predicted ncRNAs (which are outside annotated regions), their mRNA targets, 2D structures and their possible role in virulence and pathogenesis of M. tuberculosis and how they can be exploited as targets in treatments. Due tro the amount of data that is reported in this manuscript, the results and discussion sections could not be separated.
Reviewer comment
Additionally, at 2.3 section, the authors uncovered that ncRNA regulated mRNA at transcription level. Do they further detect their expression level? Namely, protein level?
Author Response
Line 476 – Line 478: the protein product of the targeted mRNA by the highly expressed ncRNA was not profiled and this remains to be performed in future studies as stated below, “Furthermore, the protein products of the targeted mRNAs will be profiled in future studies to confirm the regulatory roles of the highly expressed ncRNAs that were identified in the current study.”
Reviewer comment
Line 14: In the main text, the abbreviation for small RNA is written as “sRNA” (Line 38). Please ensure consistency throughout the manuscript.
Author Response
Line 14: smRNA Seq referrers to the specific library kit and sequencing that was performed for small RNAs.
Line 38: sRNA refers to small RNAs. smRNA is only used when we refer to sequencing while sRNA is used throughout to refer to the actual Small RNA transcripts.
Reviewer comment
Line 83: The authors stated that they used KZN strains in their analysis. Is "KZN" an abbreviation for F15/LAM4/KZN? Please clarify.
Author Response
Line 14: Yes, F15/LAM4/KZN is abbreviated as the KZN strain. It is first abbreviated in the Abstract (Line 14) and KZN is used throughout to refer to this specific Mycobacterium tuberculosis strain.
Reviewer comment
Lines 105–107: The sentence, “This was in contrast to low SDET…,” is unclear. Could the authors clarify what they are trying to demonstrate?
Author Response
Line 109 – Line 111: Minor edits have been made here to provide clarity. We were referring to the differences observed in Total RNA Seq analysis compared to the smRNA Seq analysis. These two sentences now read as follows, “The Beijing vs H37Rv strains induced the highest number of both up (630) and downregulated (465) transcripts compared to KZN vs H37Rv up (518) and downregulated (338) transcripts in Total RNA Seq reads. This was comparable to the high SDET in smRNA Seq outputs with a total of 423 and 524 transcripts in KZN vs H37Rv and Beijing vs H37Rv, respectively for both annotated and unannotated transcripts.”
Reviewer comment
Line 105: Please define what is meant by "low SDET." Are there specific criteria? If "high SDET" is also used, please define it as well.
Author Response
Line 111- Line 112: SDET was defined as “significantly differentially expressed transcripts (SDET)”. Significantly differentially expressed transcripts are transcripts that are differentially regulated between the clinical strains and the laboratory H37Rv strain. Their false discovery rate (FDR), also known as the q value is below 0.05 from the DeSeq2 analysis. This was also described in Material and Methods, Line 533 – 536 as follows, “DeSeq2 (v. 1.45.0) was used to calculate differential expression and fold changes among the KZN and Beijing clinical strain compared to the laboratory H37Rv strain using the false discovery rate (FDR) q < 0.05 (Love et al., 2014)”
Line 109 – Line 114: These two sentences have been revised. Low SDET and high SDET refers to the numbers as depicted in Table 1.
Reviewer comment
Table 1: The formatting is uneven. Please align the table properly.
Author Response
Table 1 was reformatted and aligned properly.
Reviewer comment
Lines 112–118: This section is poorly written and difficult to follow. Please revise and polish the sentences for clarity.
Author Response
Line 122 – Line 129: The sentences have been revised as follows to provide clarity as follows, “Peptidoglycan biosynthesis KEGG pathway was enriched in the common transcripts as well as in the upregulated transcripts for both the Beijing and KZN strains, versus the laboratory, H37Rv strain. Furthermore, the upregulated transcripts for both strains enriched the Microbial metabolism in diverse environment and lysine degradation KEGG pathways. Other KEGG pathways for the down-regulated transcripts were unique as shown in Figure 1D and 1-F.”
Reviewer comment
Figure 1: The font sizes within the figure are inconsistent. Please ensure uniformity.
Author Response
Figure 1 has been reformatted to ensure font sizes within the figure are consistent. Also, original images have been provided and not compressed in 1 image that result in loss of figure quality.
Reviewer comment
Lines 139–141: The content here was already mentioned in the previous paragraph. Repeating it is redundant and should be removed.
Author Response
Line 175 – Line 176: The repeated content was removed as suggested by the reviewer.
Comments on the Quality of English Language
Some sections are repetitive and overly wordy. Please revise to improve clarity and conciseness.
Author Response
The repetitive sections have been deleted, and the “overly wordy” sections have been rewritten in a more concise clear manner. The revisions and edits made within the manuscripts have been highlighted for ease of identification.
We trust that we have adequately responded to the Reviewers’ comments and queries.
Thank you for your consideration.
Kind Regards
Dr. Nontobeko Mvubu (PhD)
Senior Lecturer/Researcher
719 Umbilo Road
Doris Duke Medical Research Institute
School of Laboratory Medicine and Medical Sciences
Discipline of Medical Microbiology
College of Health Sciences
University of KwaZulu-Natal
Durban, 4000
South Africa
Email: mvubun@ukzn.ac.za
Tel: +27312601934
Reviewer 2 Report
Comments and Suggestions for Authors
The manuscript presents the results of an RNA-Seq experiment comparing three Mycobacterium tuberculosis strains: the reference laboratory strain and two clinical strains. This work addresses an important topic, and I commend the authors for their efforts in generating and analyzing this valuable dataset. I fully support the importance of making such data publicly available, as it contributes significantly to advancing our understanding of M. tuberculosis biology and its clinical implications.
Overall Appreciation:
1. While the manuscript presents an interesting dataset, I feel that the analysis remains underdeveloped, giving the impression that the work is still in its early stages. The large amount of data generated has significant potential to yield meaningful biological insights, but this has not yet been fully realized. Currently, the manuscript primarily lists the outputs of various tools without delving deeper into the analysis or biological interpretation of the results.
2. The visualization of the data could be significantly improved. This includes not only enhancing image quality (e.g., increasing resolution to avoid pixelation) but also employing more sophisticated visualizations to better represent and communicate the data. At present, the visualizations appear somewhat basic, and the lack of effort in this area does not do justice to the richness of the data collected.
3. Additionally, there are several instances where the use of English is unconventional, which can disrupt the flow and clarity of the manuscript. I recommend a thorough revision of the language to ensure smooth and effective communication of the findings.
Detailed Comments:
- Line 18-20: "In contrast, Tuberculosis and biosynthesis of siderophores and KEGG pathways were enriched by the Beijing vs H37Rv-specific transcripts." Is the word "Tuberculosis" meant to be included here? It seems unusual in this context.
- Lines 20-21: "Novel sense and antisense ncRNAs, as well as the expression of these transcripts were observed and these targeted RNA transcripts that are involved in cell wall synthesis and bacterial metabolism ion a strain-specific manner." Is the word "ion" a typo or the intended term? Please clarify.
- Lines 101-103: "A total of 856 and 1095 were significantly differentially expressed transcripts (SDET) between the KZN and Beijing strains, compared to the laboratory H37Rv, respectively from the Total RNA Seq reads." This sentence is difficult to follow due to unclear phrasing, poor sentence structure, and grammatical issues. I recommend rewriting for clarity.
- Line 103: The use of "induced" in this context may be misleading, as induction is typically associated with up-regulation in gene expression studies. Consider revising for accuracy.
- Lines 112-113: "All transcripts were enriched for KEGG pathways (Figure 1B-F) which varied among the selected transcripts." The phrasing is awkward and could benefit from clearer wording to better convey the intended meaning.
- Line 120-121: "Significantly (q < 0.05) differentially expressed transcripts (SDET) in two clinical strains of Mycobacterium tuberculosis versus the laboratory H37Rv." The word "strain" appears to be missing at the end of the sentence.
- Table 1: The differing vertical alignment in columns 2 and 3 makes the table harder to read. Consistent formatting would improve readability.
- Table 1: Repetition of "in Total RNA Seq" and "in smRNA Seq" could be avoided by indicating this information in a separate column.
- Table 1: The line "All smRNA Seq SDET" is formatted as a title line, which is confusing and inconsistent.
- Figure 1: All panels are of low resolution, making them difficult to interpret. Higher quality images are needed.
- Figure 1: None of the charts have labels on the x-axes, which hampers understanding.
- Figure 1: There is no uniformity in the use of vertical grid lines across the panels, which detracts from the overall presentation.
- Lines 126-127: "Annotated transcripts that were differentially expressed were enriched within DAVID and the 428 common transcripts were mostly involved in (B) peptidoglycan biosynthesis KEGG pathway." The meaning of "DAVID" should be clarified for readers unfamiliar with the tool.
- Lines 128-130: "Unique upregulated transcripts in Beijing vs H37Rv enriched the peptidoglycan biosynthesis, Tuberculosis, Lysine biosynthesis and microbial metabolism in diverse environments KEGG pathways." The phrasing is unorthodox and makes it unclear the direction of causality.
- Lines 132-133: "(E) Five and (F) two KEGG pathways were enriched for KZN strain compared to the laboratory strain for up- and down-regulated transcripts, respectively." Similar to the previous comment, the phrasing here is unusual and could benefit from clarification.
- Lines 143-146: "Previous studies revealed a hypervirulent phenotype of the Beijing strain in vitro [28] characterized by low cytokine response [24,29], and in vivo models [30,31] as well as its global prevalence associated with higher transmissibility [32,33] compared to other clinical strains." The phrasing is awkward and could be restructured for better readability.
- Line 149: The phrase "tuberculosis KEGG pathway" is unclear. Could the authors specify what this pathway includes or represents?
- Lines 150-153: "The presence of unique pathways in the KZN strain suggests that despite its lower number of SDETs compared to the Beijing strain, it may possess distinct mechanisms that contribute to its persistence and virulence in specific populations." Since the study was conducted in a standard laboratory growth medium, are these types of inferences valid?
- Lines 159-161: "This pathway is essential for structural support, cellular processes and virulence; hence, its enrichment is crucial in all M. tuberculosis strains [36,37]." If enrichment is observed in all strains, it would not be considered enrichment, as no differential expression would be apparent.
- Lines 171-175: "The KZN strain had the highest number of the sense (2310) and antisense (9590) RNAs from the smRNA Seq reads in contrast to the Beijing strain that exhibited the highest number of both sense (29) and antisense (370) ncRNAs in all Total RNA Seq reads, both compared to the laboratory strain, respectively (Table 2; Supplementary file 1-6)." The number of proposed small RNAs exceeds the number of genes in the organism. Is this expected? How does it compare to other organisms? Could some of these represent abortive transcriptions? How can these be distinguished from regulatory small RNAs?
- Lines 187-189: "Thus, the target mRNA transcripts for the top 5 expressed antisense RNAs were then interrogated in both smRNA and total RNA Seq data (Supplementary file 1-6)." The word "interrogated" is used in an unusual way here and could be replaced with a more conventional term.
- Lines 184-187: "Lloréns-Rico, et al. [39] stated that many of these ncRNAs expression may not induce significant changes in gene expression because their expression may be due to transcriptional noise arising at spurious promoters, thus only ncRNAs transcripts with high RPKM may have biological significance." It would be helpful to include the number of non-coding RNAs remaining after applying this criterion in Table 2 to contextualize the reported high numbers.
- Lines 198-201: "Furthermore, the smRNA Seq transcripts detected by Rockhopper included annotated protein coding genes, regulatory rRNA and tRNA and novel small RNAs as well as previously identified sRNAs such as ASdes, ASpks, AS1726 that are antisense to desA1, pks12, Rv1726 genes, respectively." The breakdown of these categories should also be included in Table 2 for clarity.
- Lines 248-250: "For each of the strains, one and three most expressed sense ncRNA was characterized by elucidating secondary structure prediction and target mRNAs together with their respective function in Total Seq and smRNA Seq, respectively." The phrasing here impairs understanding and should be restructured for clarity.
- Tables 3, 5, and 7: The "Product" column is unnecessary as it contains the same value for all transcripts.
- Tables 4, 6, and 8: The layout should be adjusted for clarity. Presenting multiple pieces of information in the same cell makes interpretation difficult.
- Tables 4, 6, and 8: A color code legend should be provided for the structural information.
- Lines 291-294: "Upon interrogation of the Rv1000c operon, only one characterized gene was identified as a Probable arginine deiminase ArcA (arginine dihydrolase) known as Rv1001, while the rest of the genes (Rv0999, Rv0998, Rv0997, Rv1002c) encoded hypothetical proteins." The word "interrogation" is unconventional here and could be replaced with "analysis."
- Figure 2: In panel B, the color coding is counterintuitive, with red indicating high expression and green indicating low expression. Additionally, scales are not labeled with their units. It is surprising that the visually dissimilar strains cluster together—could this be explained?
- Figure 2: In panel C, a legend for the color coding is needed.
- Figure 2: All visuals are of low resolution, which makes interpretation difficult. Higher-quality images are necessary.
- Lines 413-421: Non-default parameters for each tool should be explicitly stated. If only default parameters were used, this should also be mentioned.
- Lines 423-426: "Rockhopper (v2.03) was used to predict ncRNAs (sRNA and long non coding RNAs) together with their genomic locations and their respective operons using the reference based analysis from well-annotated strains of M. tuberculosis [81] from trimmed RNA Seq fastq files." The phrasing is unusual and hampers understanding. I suggest rewording for clarity.
- Lines 428-430: Accession numbers should be provided instead of hyperlinks for clarity and standardization.
- Lines 430-432: "Rockhopper interrogated noncoding RNAs using strand-guided parameters to reveal the genomic location and strand orientation of the identified noncoding RNAs." There is a typo in this sentence that should be corrected.
- Lines 474-481: The instructions for authors should be removed from this section.
- Lines 484-486: Similarly, the instructions for authors in this section should be removed.
Concluding remarks:
I appreciate the authors’ efforts in generating this valuable RNA Seq dataset and recognize the importance of making this data accessible to the scientific community. The work has the potential to contribute meaningfully to our understanding of Mycobacterium tuberculosis transcriptomics. However, as it stands, the analysis, visualization, and interpretation of the data leave significant room for improvement.
To unlock the full potential of this dataset, more attention needs to be directed toward exploring the data in greater depth, employing advanced visualization techniques, and deriving meaningful biological insights. Currently, the manuscript largely presents a list of RNAs, which, while informative, could benefit from a more thorough analysis to provide a deeper understanding of the biological processes and mechanisms involved.
I hope that my suggestions, along with the detailed feedback provided, will support the authors in further developing their manuscript. Additionally, I encourage the authors to consult recent RNA Seq studies that delve deeper into analysis and interpretation, as these may serve as valuable guides for enhancing their approach.
I look forward to seeing how this work evolves and the contributions it can make to the field. Thank you for sharing your research.
Comments on the Quality of English LanguageWhile I am not a native English speaker, I have worked in science for several years and have had the opportunity to live and work abroad, which has allowed me to become a proficient user of the language. With this perspective, I noticed that the manuscript contains some unusual word choices and phrasing that affect the flow of the text and the clarity of the message the authors are trying to convey.
Improving the phrasing and grammatical structure would greatly enhance the readability and impact of the manuscript. I strongly recommend that the authors rephrase and revise it to ensure their ideas are communicated more effectively. Fortunately, there are many online tools available today that can assist with English grammar and style. Leveraging these tools could be particularly helpful during the revision process.
Clear communication is key to maximizing the impact of this important work, and I hope these suggestions support the authors in achieving that goal.
Author Response
College of Health Sciences
School of Laboratory Medicine and Medical Sciences
Medical Microbiology
Dr. Nicole Hansmeier
Editor-in-Chief
International Journal of Molecular Sciences
Dear Dr. Nicole Hansmeier
11 December 2024
Submission of manuscript Revision to International Journal of Molecular Sciences: Comparative transcriptomics reveal differential expression of coding and non-coding RNAs in clinical strains of Mycobacterium tuberculosis
We thank the Reviewers for their valuable comments. The queries and comments requested by reviewers are addressed in the point-by-point response below and highlighted in the manuscript file. Due to changes during manuscript edits, initial line numbers have changed.
Reviewers Comments
Reviewer 2
Reviewer comment
The manuscript presents the results of an RNA-Seq experiment comparing three Mycobacterium tuberculosis strains: the reference laboratory strain and two clinical strains. This work addresses an important topic, and I commend the authors for their efforts in generating and analyzing this valuable dataset. I fully support the importance of making such data publicly available, as it contributes significantly to advancing our understanding of M. tuberculosis biology and its clinical implications.
Author Response
We thank the Reviewer for the appreciation of the work presented in this manuscript.
Reviewer comment
Overall Appreciation:
While the manuscript presents an interesting dataset, I feel that the analysis remains underdeveloped, giving the impression that the work is still in its early stages. The large amount of data generated has significant potential to yield meaningful biological insights, but this has not yet been fully realized. Currently, the manuscript primarily lists the outputs of various tools without delving deeper into the analysis or biological interpretation of the results.
The visualization of the data could be significantly improved. This includes not only enhancing image quality (e.g., increasing resolution to avoid pixelation) but also employing more sophisticated visualizations to better represent and communicate the data. At present, the visualizations appear somewhat basic, and the lack of effort in this area does not do justice to the richness of the data collected.
Additionally, there are several instances where the use of English is unconventional, which can disrupt the flow and clarity of the manuscript. I recommend a thorough revision of the language to ensure smooth and effective communication of the findings.
Author Response
The tools and figures generated within the manuscripts have been utilized by previous studies that report on similar work as follows, DOIs: 10.1007/s11274-024-04089-6 and 10.1007/s00284-021-02733-0
Despite the progress made in sequencing technologies in the last decade, investigating the role of non-coding RNAs has been very slow in M. tuberculosis research. Once all the non-coding RNAs are identified within all lineages of the Mycobacterium tuberculosis complex, the database should be created (work that is in the pipeline) to showcase the molecular networks between the ncRNAs and their respective targets as well as their associated pathways (and how these networks and pathways contribute to M. tuberculosis virulence).
We have added new data (Figure 2 and Figure 3) to shown molecular networks for protein-protein interaction for the KZN Transcripts 161 and Beijing Transcript 345 mRNA targets. This new data is described in Line 299 – 301; Line 305 - 309 and discussed in Line 403 – 412. Description of how the networks were generated is provided in Line 569 – 571.
Lastly, we have proofread the manuscript and made changes where necessary (as highlighted in the document) to improve the English and readability of the work.
Reviewer comment
Line 18-20: "In contrast, Tuberculosis and biosynthesis of siderophores and KEGG pathways were enriched by the Beijing vs H37Rv-specific transcripts." Is the word "Tuberculosis" meant to be included here? It seems unusual in this context.
Author Response
Line 18 – Line 20: Yes, Tuberculosis is meant to be here. It is one of the KEGG pathways that was enriched but there was a typo here of “and” and now the sentence reads, “In contrast, Tuberculosis and biosynthesis of siderophores KEGG pathways were enriched by the Beijing vs H37Rv-specific transcripts.”
Reviewer comment
Lines 20-21: "Novel sense and antisense ncRNAs, as well as the expression of these transcripts were observed and these targeted RNA transcripts that are involved in cell wall synthesis and bacterial metabolism ion a strain-specific manner." Is the word "ion" a typo or the intended term? Please clarify.
Author Response
Line 20 – Line 22: The “ion” word was a typo. It has been corrected and now reads, “Novel sense and antisense ncRNAs, as well as the expression of these transcripts were observed and these targeted RNA transcripts that are involved in cell wall synthesis and bacterial metabolism in a strain-specific manner.”
Reviewer comment
Lines 101-103: "A total of 856 and 1095 were significantly differentially expressed transcripts (SDET) between the KZN and Beijing strains, compared to the laboratory H37Rv, respectively from the Total RNA Seq reads." This sentence is difficult to follow due to unclear phrasing, poor sentence structure, and grammatical issues. I recommend rewriting for clarity.
Author Response
Line 106 – Line 109: The sentence has been revised as follows, “DeSeq2 analysis revealed a total of 856 and 1095 significantly differentially expressed transcripts (SDET) between the KZN and Beijing strains, compared to the laboratory H37Rv, respectively from the Total RNA Seq reads”
Reviewer comment
Line 103: The use of "induced" in this context may be misleading, as induction is typically associated with up-regulation in gene expression studies. Consider revising for accuracy.
Author Response
Line 109 – Line 111: Induced was changed to exhibited and the sentence now reads as follows, “The Beijing vs H37Rv strains exhibited the highest number of both up (630) and downregulated (465) transcripts compared to KZN vs H37Rv up (518) and downregulated (338) transcripts in Total RNA Seq reads.”
Reviewer comment
Lines 112-113: "All transcripts were enriched for KEGG pathways (Figure 1B-F) which varied among the selected transcripts." The phrasing is awkward and could benefit from clearer wording to better convey the intended meaning.
Author Response
Line 120 – Line 121: The sentence has been revised as follows, “The up- and down-regulated SDET among the strains enriched various KEGG pathways (Figure 1B-F).”
Reviewer comment
Line 120-121: "Significantly (q < 0.05) differentially expressed transcripts (SDET) in two clinical strains of Mycobacterium tuberculosis versus the laboratory H37Rv." The word "strain" appears to be missing at the end of the sentence.
Author Response
Line 131 – Line 132: Table 1 heading has been corrected by adding strain at the end of the sentence as follows, “Table 1. Significantly (q < 0.05) differentially expressed transcripts (SDET) in two clinical strains of Mycobacterium tuberculosis versus the laboratory H37Rv strain.”
Reviewer comment
Table 1: The differing vertical alignment in columns 2 and 3 makes the table harder to read. Consistent formatting would improve readability.
Author Response
Table 1 was reformatted as follows to improve readability as shown below:
Table 1. Significantly (q < 0.05) differentially expressed transcripts (SDET) in two clinical strains of Mycobacterium tuberculosis versus the laboratory H37Rv strain.
KZN vs H37Rv |
Beijing vs H37Rv |
|
Total RNA Seq All SDET |
856 |
1095 |
Upregulated known transcripts |
257 |
308 |
Upregulated unknown transcripts |
261 |
322 |
Downregulated known transcripts |
158 |
219 |
Downregulated unknown transcripts |
180 |
246 |
smRNA Seq All SDET |
423 |
524 |
Upregulated known transcripts |
111 |
169 |
Upregulated unknown transcripts |
110 |
143 |
Downregulated known transcripts |
99 |
93 |
Downregulated unknown transcripts |
103 |
119 |
Reviewer comment
Table 1: Repetition of "in Total RNA Seq" and "in smRNA Seq" could be avoided by indicating this information in a separate column.
Author Response
Table 1: Two additional rows were added to avoid repetition of Total RNA Seq and smRNA Seq in each row as follows:
KZN vs H37Rv |
Beijing vs H37Rv |
|
Total RNA Seq All SDET |
856 |
1095 |
Upregulated known transcripts |
257 |
308 |
Upregulated unknown transcripts |
261 |
322 |
Downregulated known transcripts |
158 |
219 |
Downregulated unknown transcripts |
180 |
246 |
smRNA Seq All SDET |
423 |
524 |
Upregulated known transcripts |
111 |
169 |
Upregulated unknown transcripts |
110 |
143 |
Downregulated known transcripts |
99 |
93 |
Downregulated unknown transcripts |
103 |
119 |
Reviewer comment
Table 1: The line "All smRNA Seq SDET" is formatted as a title line, which is confusing and inconsistent.
Author Response
Table 1 has been reformatted to avoid confusion as follows:
KZN vs H37Rv |
Beijing vs H37Rv |
|
Total RNA Seq All SDET |
856 |
1095 |
Upregulated known transcripts |
257 |
308 |
Upregulated unknown transcripts |
261 |
322 |
Downregulated known transcripts |
158 |
219 |
Downregulated unknown transcripts |
180 |
246 |
smRNA Seq All SDET |
423 |
524 |
Upregulated known transcripts |
111 |
169 |
Upregulated unknown transcripts |
110 |
143 |
Downregulated known transcripts |
99 |
93 |
Downregulated unknown transcripts |
103 |
119 |
Reviewer comment
Figure 1: All panels are of low resolution, making them difficult to interpret. Higher quality images are needed.
Author Response
Figure 1: The original images for Figure 1 were provided without being formatted to improve quality and text within figure. Furthermore, the font text within each panel was increased (to 14) and text heading was bolded.
Reviewer comment
Figure 1: None of the charts have labels on the x-axes, which hampers understanding.
Author Response
Figure 1: X and Y axis have been inserted and labelled to improve the understanding of the figures.
Reviewer comment
Figure 1: There is no uniformity in the use of vertical grid lines across the panels, which detracts from the overall presentation.
Author Response
Figure 1: All vertical lines within the panels have been removed in the new reformatted images.
Reviewer comment
Lines 126-127: "Annotated transcripts that were differentially expressed were enriched within DAVID and the 428 common transcripts were mostly involved in (B) peptidoglycan biosynthesis KEGG pathway." The meaning of "DAVID" should be clarified for readers unfamiliar with the tool.
Author Response
Line 159 – Line 160: DAVID has been defined and this sentence is as follows, “Annotated transcripts that were differentially expressed were enriched within the Database for Annotation, Visualization, and Integrated Discovery (DAVID) and the 428 common transcripts were mostly involved in (B) peptidoglycan biosynthesis KEGG pathway”.
Reviewer comment
Lines 128-130: "Unique upregulated transcripts in Beijing vs H37Rv enriched the peptidoglycan biosynthesis, Tuberculosis, Lysine biosynthesis and microbial metabolism in diverse environments KEGG pathways." The phrasing is unorthodox and makes it unclear the direction of causality.
Author Response
Line 161 – Line 164: This sentence has been revised as follows, “Beijing vs H37Rv-specific upregulated the following SDET enriched KEGG pathways, peptidoglycan biosynthesis, Tuberculosis, Lysine biosynthesis and microbial metabolism in diverse environments”
Reviewer comment
Lines 132-133: "(E) Five and (F) two KEGG pathways were enriched for KZN strain compared to the laboratory strain for up- and down-regulated transcripts, respectively." Similar to the previous comment, the phrasing here is unusual and could benefit from clarification.
Author Response
Lines 165 – Line 168: Specific pathways have been mentioned to provide clarity in this sentence as follows, “(E) Peptidoglycan biosynthesis, Microbial metabolism in diverse environments, Steroid degradation, Lysine biosynthesis and Fatty acid degradation; as well as the (F) Bacterial secretion system and Fructose and mannose metabolism KEGG pathways were enriched for up- and down-regulated transcripts, respectively, for KZN strain compared to the laboratory strain”.
Reviewer comment
Lines 143-146: "Previous studies revealed a hypervirulent phenotype of the Beijing strain in vitro [28] characterized by low cytokine response [24,29], and in vivo models [30,31] as well as its global prevalence associated with higher transmissibility [32,33] compared to other clinical strains." The phrasing is awkward and could be restructured for better readability.
Author Response
Line 179 – Line 183: This sentences has been rephrased and separated into two sentences as follows, “Previous studies revealed a hypervirulent phenotype of the Beijing strain, that is characterized by low cytokine responses (Wang et al., 2010, Mvubu et al., 2018),in in vitro (Ribeiro et al., 2014) and in vivo infection models (Bespyatykh et al., 2019, Vinogradova et al., 2021). Additionally, its global prevalence is associated with high transmissibility (Luo et al., 2022, Zhu et al., 2023) compared to other clinical strains.”
Reviewer comment
Line 149: The phrase "tuberculosis KEGG pathway" is unclear. Could the authors specify what this pathway includes or represents?
Author Response
Line 186 – Line 189: Tuberculosis KEGG pathway is the enrichment of the pathway of bacterial (M. tuberculosis) and some of the host genes (macrophages and dendritic cells) that play a critical role in TB pathogenesis. It is designated as mtu05152:Tuberculosis and the actual pathway is shown here: https://www.genome.jp/pathway/mtu05152. This has been described in the manuscript as follows, “The current study also revealed that the Beijing strain uniquely upregulated transcripts involved in Tuberculosis KEGG pathway, which include pathogen- and host-specific transcript that play critical role in pathogenesis, while the KZN strain upregulated transcripts were involved in fatty acid and steroid degradation KEGG pathways”
Reviewer comment
Lines 150-153: "The presence of unique pathways in the KZN strain suggests that despite its lower number of SDETs compared to the Beijing strain, it may possess distinct mechanisms that contribute to its persistence and virulence in specific populations." Since the study was conducted in a standard laboratory growth medium, are these types of inferences valid?
Author Response
Line 190 – Line 193: This sentence has been revised: “The presence of uniquely enriched KEGG pathways in the KZN strain suggests that despite its lower number of SDETs compared to the Beijing strain, it may possess distinct mechanisms that contribute to its persistence and virulence in specific populations, and this remains to be investigated in genetically diverse strains and infected individuals.” It must be stated that there is scarcity of information on host-specific and pathogen-specific transcripts that are used by genetically diverse strains to evade elimination by host defences.
Reviewer comment
Lines 159-161: "This pathway is essential for structural support, cellular processes and virulence; hence, its enrichment is crucial in all M. tuberculosis strains [36,37]." If enrichment is observed in all strains, it would not be considered enrichment, as no differential expression would be apparent.
Author Response
Line 199 – Line 203: Enrichment within pathways means that we have detected transcripts that play a significant role within that pathway. In this case, different transcripts (which are differentially regulated within the clinical strains vs H37Rv strain) enriched the same pathway (peptidoglycan biosynthesis pathway). There was a significant differential (q < 0.05) expression of these transcripts (they are not the same transcripts) and they all enriched this pathway (specific transcripts in different part of the pathway).
Reviewer comment
Lines 171-175: "The KZN strain had the highest number of the sense (2310) and antisense (9590) RNAs from the smRNA Seq reads in contrast to the Beijing strain that exhibited the highest number of both sense (29) and antisense (370) ncRNAs in all Total RNA Seq reads, both compared to the laboratory strain, respectively (Table 2; Supplementary file 1-6)." The number of proposed small RNAs exceeds the number of genes in the organism. Is this expected? How does it compare to other organisms? Could some of these represent abortive transcriptions? How can these be distinguished from regulatory small RNAs?
Author Response
The number of proposed ncRNA does exceed the number of protein coding genes as shown in Table 2. This was expected due to not applying cutoff values to provide an overall understanding of how many non-coding RNAs are present in each strain of M. tuberculosis used in this study (that was compared to the laboratory strain). The only study that used smRNA Seq to elucidate ncRNA presence and expression (https://link.springer.com/article/10.1007/s11274-024-04089-6) detected 351 sRNAs after applying several filters as described in their material and methods as foolows, “Default parameters were used, and the predicted sRNAs were ranked by the average coverage among the replications. The average coverage scores equal to or greater than 30 was used as the cut-off value to remove lowly expressed sRNAs.” It should be stated that the current study also applied high sequencing depth of 20 million reads for smRNA Seq (mentioned in Line 522 of the manuscript) and this information was not provided by Jumat and Chin (2024) recent study (which could explain our ability to detect a very high number of ncRNAs). All the transcripts (coding, non-coding as well as abortive transcripts) will be detected by smRNA Seq, however, for the purpose of the current manuscript, only regulatory ncRNAs and their targets are presented. Lastly, M. tuberculosis has ~4000 genes that can be regulated by multiple cis- and trans-encoded ncRNAs. This was the case for many antisense ncRNAs as shown in Supplementary file 1-6 with their respective targets.
Reviewer comment
Lines 187-189: "Thus, the target mRNA transcripts for the top 5 expressed antisense RNAs were then interrogated in both smRNA and total RNA Seq data (Supplementary file 1-6)." The word "interrogated" is used in an unusual way here and could be replaced with a more conventional term.
Author Response
Line 229 – Line 231: “Interrogated” was replaced with “investigated” and the sentence now reads as follows, “Thus, the target mRNA transcripts for the top 5 expressed antisense RNAs were investigated in both smRNA and total RNA Seq data (Supplementary file 1-6)”.
Reviewer comment
Lines 184-187: "Lloréns-Rico, et al. [39] stated that many of these ncRNAs expression may not induce significant changes in gene expression because their expression may be due to transcriptional noise arising at spurious promoters, thus only ncRNAs transcripts with high RPKM may have biological significance." It would be helpful to include the number of non-coding RNAs remaining after applying this criterion in Table 2 to contextualize the reported high numbers.
Author Response
Table 2 represents the overall findings for all strains after using Rockhopper. The 10 highest expressed ncRNAs (following Lloréns-Rico, et al. [39] that indicated that only highly expressed ncRNAs will have an impact on their mRNA targets) were selected and presented in Table 3, Table 5 and Table 7. We provided supplementary material (Supplementary file 1-6) to show all the ncRNAs that we detected as they may serve as an important template for future Mycobacterium tuberculosis studies in this area since there’s very limited literature on ncRNAs for this pathogen, especially for genetically diverse clinical strains.
Reviewer comment
Lines 198-201: "Furthermore, the smRNA Seq transcripts detected by Rockhopper included annotated protein coding genes, regulatory rRNA and tRNA and novel small RNAs as well as previously identified sRNAs such as ASdes, ASpks, AS1726 that are antisense to desA1, pks12, Rv1726 genes, respectively." The breakdown of these categories should also be included in Table 2 for clarity.
Author Response
Table 2 has been revised as follows to add tRNA, protein coding transcripts and sense as well as antisense RNA as previously demonstrated. We did not pick up any rRNA transcripts (due to Ribozero depletion performed during library preparation). We used Rockhopper to investigate regulatory (sense and antisense) RNAs and their targets and not protein coding transcripts as these have been enriched in Figure 1 using the DeSeq2 tool. Hence, we initially were not interested in these transcripts for this section. Also, we used the same annotated whole genome sequence for each clinical strain, hence, protein coding transcripts remained the same for smRNA Seq and Total RNA Seq (we do suggest that we remove the highlighted sections below).
|
KZN |
Beijing |
H37Rv |
Total RNA Seq |
|
|
|
5'UTRs |
47 |
38 |
516 |
3'UTRs |
133 |
112 |
569 |
Sense predicted RNAs |
10 |
29 |
13 |
Antisense predicted RNAs |
195 |
370 |
244 |
Annotated protein coding transcripts |
2426 |
2497 |
3178 |
Hypothetical proteins transcripts |
1578 |
1616 |
861 |
rRNA |
0 |
0 |
0 |
tRNA |
45 |
45 |
45 |
|
|
|
|
smRNA Seq |
|
|
|
5'UTRs |
926 |
888 |
893 |
3'UTRs |
751 |
721 |
722 |
Sense predicted RNAs |
2310 |
2154 |
2193 |
Antisense predicted RNAs |
9590 |
4880 |
8241 |
Annotated protein coding transcripts |
2426 |
2497 |
3178 |
Hypothetical proteins transcripts |
1578 |
1616 |
861 |
rRNA |
0 |
0 |
0 |
tRNA |
45 |
45 |
45 |
Reviewer comment
Lines 248-250: "For each of the strains, one and three most expressed sense ncRNA was characterized by elucidating secondary structure prediction and target mRNAs together with their respective function in Total Seq and smRNA Seq, respectively." The phrasing here impairs understanding and should be restructured for clarity.
Author Response
Line 293 – Line 298: This sentence has been rephrased as follows, “The three most expressed ncRNAs from smRNA Seq and the one highly expressed ncRNA from Total Seq were characterized by elucidating their predicted secondary structures, target mRNAs as well as target mRNA functions.”.
Reviewer comment
Tables 3, 5, and 7: The "Product" column is unnecessary as it contains the same value for all transcripts.
Author Response
Table 3, Table 5 and Table 7: The product column has been deleted in all three Tables.
Reviewer comment
Tables 4, 6, and 8: The layout should be adjusted for clarity. Presenting multiple pieces of information in the same cell makes interpretation difficult.
Author Response
Table 4, Table 6 and Table 8: Respectfully, the formatting in these Tables has not been changed because this is the most efficient way to provide ncRNA structure, function, their targets and the binding probability of their target. The recent study presenting similar data (shown in their Table 4) for M. tuberculosis Lineage 1 ncRNAs (https://link.springer.com/article/10.1007/s11274-024-04089-6) used the same format to present their data.
Reviewer comment
Tables 4, 6, and 8: A color code legend should be provided for the structural information.
Author Response
Table 4, Table 6, and Table 8: A footer with color coding for the ncRNA structures has been provided as follows: *: ncRNA structure color coding
Reviewer comment
Lines 291-294: "Upon interrogation of the Rv1000c operon, only one characterized gene was identified as a Probable arginine deiminase ArcA (arginine dihydrolase) known as Rv1001, while the rest of the genes (Rv0999, Rv0998, Rv0997, Rv1002c) encoded hypothetical proteins." The word "interrogation" is unconventional here and could be replaced with "analysis."
Author Response
Line 383 – Line 386: The word interrogation was replaced with analysis as follows, “Upon analysis of the Rv1000c operon, only one characterized gene was identified as a Probable arginine deiminase ArcA (arginine dihydrolase) known as Rv1001, while the rest of the genes (Rv0999, Rv0998, Rv0997, Rv1002c) encoded hypothetical proteins”.
Reviewer comment
Figure 2: In panel B, the color coding is counterintuitive, with red indicating high expression and green indicating low expression. Additionally, scales are not labeled with their units. It is surprising that the visually dissimilar strains cluster together—could this be explained?
Author Response
Figure 4 (Which was the original Figure 2): Most heatmaps presented in literature use red to indicate high expression and green or blue to represent low expression. Please see few studies here: DOI: 10.1242/dmm.018366; 10.1155/2014/986048; 10.1186/s13029-014-0030-2; 10.1093/bioinformatics/btw313. The scales were represented with their units which was 0, 20 and 107 (as these represented RPKM values of expression). It was not surprising that Beijing and H37Rv clustered together because their expression profiles were closely related compared to very low expression profiles observed for the KZN strain. Supplementary Table 3 below has original values that were used to make this Heat Map and it is clear that KZN strain is an outlier compared to the Beijing and H37Rv strains.
Supplementary Table 3: The 36 common differentially expressed sRNAs in clinical strains of M. tuberculosis.
KZN |
Beijing |
H37Rv |
|
Transcript_32 |
4 |
15 |
151 |
Transcript_15 |
0 |
16 |
72 |
Transcript_34 |
74 |
67 |
0 |
Transcript_36 |
0 |
19 |
91 |
Transcript_16 |
4 |
35 |
156 |
Transcript_20 |
0 |
20 |
131 |
Transcript_21 |
0 |
11 |
141 |
Transcript_129 |
0 |
6 |
82 |
Transcript_46 |
0 |
7 |
104 |
Transcript_39 |
0 |
14 |
89 |
Transcript_44 |
0 |
15 |
61 |
Transcript_20 |
0 |
19 |
66 |
Transcript_11 |
0 |
2 |
80 |
Transcript_42 |
3 |
6 |
87 |
Transcript_9 |
0 |
0 |
48 |
Transcript_48 |
12 |
3 |
152 |
Transcript_41 |
10 |
20 |
116 |
Transcript_43 |
22 |
32 |
217 |
Transcript_19 |
153 |
165 |
18 |
Transcript_44 |
9 |
13 |
81 |
Transcript_18 |
16 |
19 |
132 |
Transcript_79 |
11 |
9 |
88 |
Transcript_95 |
10 |
6 |
79 |
Transcript_33 |
41 |
9 |
256 |
Transcript_18 |
95 |
62 |
16 |
Transcript_19 |
19 |
6 |
109 |
Transcript_63 |
50 |
79 |
254 |
Transcript_172 |
107 |
72 |
353 |
Transcript_14 |
0 |
159 |
531 |
Transcript_130 |
0 |
1770 |
6092 |
Transcript_179 |
0 |
278 |
661 |
Transcript_42 |
0 |
1 |
74 |
Transcript_24 |
0 |
6 |
60 |
Transcript_14 |
0 |
26 |
103 |
Transcript_215 |
0 |
51 |
147 |
Transcript_37 |
0 |
10 |
86 |
Reviewer comment
Figure 2: In panel C, a legend for the color coding is needed.
Author Response
Figure 4: The colour coding for Panel C has been added.
Reviewer comment
Figure 2: All visuals are of low resolution, which makes interpretation difficult. Higher-quality images are necessary.
Author Response
Figure 2: The original figures (unformatted) have been provided to improve the quality of each figure panel.
Reviewer comment
Lines 413-421: Non-default parameters for each tool should be explicitly stated. If only default parameters were used, this should also be mentioned.
Author Response
Line 540 – Line 542: only Trimmomatic was used with Nextera adapters option selected to trim off the sequencing adapters. Other tools were used with default options. This has been mentioned as follows, “Nextera adapter option was turned on for Trimmomatic, while default parameters were used for HiSat2, StringTie and DeSeq2 bioinformatics tools.”.
Reviewer comment
Lines 423-426: "Rockhopper (v2.03) was used to predict ncRNAs (sRNA and long noncoding RNAs) together with their genomic locations and their respective operons using the reference-based analysis from well-annotated strains of M. tuberculosis [81] from trimmed RNA Seq fastq files." The phrasing is unusual and hampers understanding. I suggest rewording for clarity.
Author Response
Line 545 – Line 547: This sentence was rephrased as follows, “Rockhopper (v2.03) used reference-based analysis from well-annotated strains of M. tuberculosis to predict ncRNAs (sRNA and long noncoding RNAs), together with their genomic locations as well as their respective operons from trimmed RNA Seq fastq files”.
Reviewer comment
Lines 428-430: Accession numbers should be provided instead of hyperlinks for clarity and standardization.
Author Response
Line 550 – Line 553: The hyperlinks have been deleted and the sentence was rephrased as follows, “M. tuberculosis H37Rv (CP003248.2), M. tuberculosis str. Beijing/NITR203 (478739819) and M. tuberculosis KZN605 (392051818) sequences downloaded by Rockhopper from RefSeq using their accession numbers were used as background references for each strain”.
Reviewer comment
Lines 430-432: "Rockhopper interrogated noncoding RNAs using strand-guided parameters to reveal the genomic location and strand orientation of the identified noncoding RNAs." There is a typo in this sentence that should be corrected.
Author Response
Line 555 – Line 557: This sentence was rephrased as follows, “Rockhopper used strand-guided parameters to reveal the genomic location and strand orientation of the identified noncoding RNAs”.
Reviewer comment
Lines 474-481: The instructions for authors should be removed from this section.
Author Response
Line 602- Line 609: Instruction for authors was deleted in this section.
Reviewer comment
Lines 484-486: Similarly, the instructions for authors in this section should be removed.
Author Response
Line 612 – 614: Instruction for authors was deleted in this section.
Reviewer comment
Concluding remarks:
I appreciate the authors’ efforts in generating this valuable RNA Seq dataset and recognize the importance of making this data accessible to the scientific community. The work has the potential to contribute meaningfully to our understanding of Mycobacterium tuberculosis transcriptomics. However, as it stands, the analysis, visualization, and interpretation of the data leave significant room for improvement.
To unlock the full potential of this dataset, more attention needs to be directed toward exploring the data in greater depth, employing advanced visualization techniques, and deriving meaningful biological insights. Currently, the manuscript largely presents a list of RNAs, which, while informative, could benefit from a more thorough analysis to provide a deeper understanding of the biological processes and mechanisms involved.
I hope that my suggestions, along with the detailed feedback provided, will support the authors in further developing their manuscript. Additionally, I encourage the authors to consult recent RNA Seq studies that delve deeper into analysis and interpretation, as these may serve as valuable guides for enhancing their approach.
I look forward to seeing how this work evolves and the contributions it can make to the field. Thank you for sharing your research.
Author Response
We have addressed all the Reviewer’s comments and queries as detailed above. Furthermore, we have added mRNA targets networks to improve our understanding of biological insights of the role of ncRNAs in M. tuberculosis. Despite progress made in RNA Seq, investigating ncRNAs and their associated networks has been only limited to their mRNA targets and the role of these targerted mRNAs (which has been presented in this manuscript). This work serves as a template of all ncRNAs in Lineage 2 and Lineage 4 of the Mycobacterium tuberculosis complex and once they have been identified in other lineages (ongoing work within the research group), we will create a database to link the known molecular networks and pathways to these ncRNAs and the expression of these transcripts under different biological conditions.
Reviewer comment
Comments on the Quality of English Language
While I am not a native English speaker, I have worked in science for several years and have had the opportunity to live and work abroad, which has allowed me to become a proficient user of the language. With this perspective, I noticed that the manuscript contains some unusual word choices and phrasing that affect the flow of the text and the clarity of the message the authors are trying to convey.
Improving the phrasing and grammatical structure would greatly enhance the readability and impact of the manuscript. I strongly recommend that the authors rephrase and revise it to ensure their ideas are communicated more effectively. Fortunately, there are many online tools available today that can assist with English grammar and style. Leveraging these tools could be particularly helpful during the revision process.
Clear communication is key to maximizing the impact of this important work, and I hope these suggestions support the authors in achieving that goal.
Author Response
We have rephrased/restructured the unclear sentences/phrases as pointed out by the reviewer for improved clarity. We appreciate the Reviewer’s extremely useful suggestions to improve the manuscript.
We trust that we have adequately responded to the Reviewers’ comments and queries.
Thank you for your consideration.
Kind Regards
Dr. Nontobeko Mvubu (PhD)
Senior Lecturer/Researcher
719 Umbilo Road
Doris Duke Medical Research Institute
School of Laboratory Medicine and Medical Sciences
Discipline of Medical Microbiology
College of Health Sciences
University of KwaZulu-Natal
Durban, 4000
South Africa
Email: mvubun@ukzn.ac.za
Tel: +27312601934
Round 2
Reviewer 2 Report
Comments and Suggestions for Authors
I would like to commend the authors for their significant efforts in addressing the reviewers' comments. It is evident that considerable thought and work went into revising the manuscript, and I believe these efforts have paid off. The quality of the manuscript has improved substantially. This revised version is a stronger contribution to the field, and I appreciate the authors’ diligence in responding to the feedback provided.
In my initial review, I expressed several major concerns regarding the manuscript. Specifically, I noted that the data analysis appeared underdeveloped, with limited biological insights being drawn from the results. The visual presentation of the data was also a significant issue, as figures lacked clarity and quality, and the visualizations did not effectively communicate the findings. Additionally, the language and phrasing in many sections of the manuscript were unusual and at times unclear, which hindered the readability and overall flow. These aspects collectively limited the impact and interpretability of the study in its original form.
I am pleased to note that the issues related to the use of the English language have been thoroughly addressed in this revised version. The text now flows much more smoothly, and the phrasing is now cleared. I congratulate the authors on this improvement, as it greatly enhances the manuscript's readability. Regarding the biological interpretation of the data, I see a noticeable improvement in this aspect as well. While challenges remain in making biological sense of all the generated data, I appreciate the authors' efforts to provide additional context and their arguments highlighting the inherent difficulties in this area. These revisions demonstrate thoughtful engagement with the feedback provided.
However, I still feel that the quality of the images remains a significant issue. The current visualizations do not do justice to the quality of the work and, unfortunately, detract from the overall impact of the manuscript. The use of print screens from analysis tools not only reflects a lack of effort in improving the presentation but also raises potential copyright concerns, which should be addressed by the journal. While the authors reference other studies to justify the use of certain types of visualizations (e.g., bar charts, Venn diagrams, and network charts), I believe the quality of these visualizations in their manuscript falls short in comparison. While the types of graphs are similar, they do not approach the same standard in terms of clarity, detail, or overall presentation. As this is a point of disagreement between the authors and myself, I will leave it to the editor to assess and make a final decision.
Detailed comments:
Line 50: There is an extra comma after the word "complex." Please consider removing it.
Line 111: One extra space is found in this line. Kindly remove it for consistency.
Line 123: One extra space is found in this line. Please correct this formatting issue.
Line 171: One extra space is found in this line. It would be great to fix this to ensure uniformity.
Table 2: There seems to be multiple font types or sizes used. It would enhance the presentation if the same font type and size were used consistently throughout the table.
Line 298: There is one extra space, and a sentence starts without a capital letter. Please correct this for proper grammar and formatting.
Figure 2: This figure contains print screens of an online tool. I recommend verifying the copyright for these images. Additionally, the meaning of FDR should be included in the figure legend for clarity.
Figure 3: This figure also contains print screens of an online tool. Please verify the copyright status. It would be helpful to include an explanation of the FDR in the figure legend.
Figure 4: This figure contains print screens of an online tool. Kindly verify the copyright for these images.
Regarding the STRING print screens: What kind of biological insights can one derive from a network where most genes are identified only by their locus, and functions are not attributed? One of the references the authors mention provides a more informative network with annotated functions highlighted within the network, offering clearer biological context. I believe this approach would strengthen the manuscript.
For all figures and tables where "ncRNA structure color coding" is used, I would suggest explaining in the legend what the color coding represents, such as whether it encodes probability or confidence levels for the proposed structure. This clarification will help readers better interpret the visualized data.
Author Response
Dr. Nicole Hansmeier
Special issue: Editor-in-Chief (Microbial Omics)
International Journal of Molecular Sciences
Dear Dr. Nicole Hansmeier
17 December 2024
Submission of manuscript Revision to International Journal of Molecular Sciences: Comparative transcriptomics reveal differential expression of coding and non-coding RNAs in clinical strains of Mycobacterium tuberculosis
We thank the Reviewers for their valuable comments. The queries and comments requested by reviewers are addressed in the point-by-point response below and highlighted in the manuscript file. Due to changes during manuscript edits, initial line numbers have changed.
Reviewers Comments
Reviewer comment
I would like to commend the authors for their significant efforts in addressing the reviewers' comments. It is evident that considerable thought and work went into revising the manuscript, and I believe these efforts have paid off. The quality of the manuscript has improved substantially. This revised version is a stronger contribution to the field, and I appreciate the authors’ diligence in responding to the feedback provided.
In my initial review, I expressed several major concerns regarding the manuscript. Specifically, I noted that the data analysis appeared underdeveloped, with limited biological insights being drawn from the results. The visual presentation of the data was also a significant issue, as figures lacked clarity and quality, and the visualizations did not effectively communicate the findings. Additionally, the language and phrasing in many sections of the manuscript were unusual and at times unclear, which hindered the readability and overall flow. These aspects collectively limited the impact and interpretability of the study in its original form.
I am pleased to note that the issues related to the use of the English language have been thoroughly addressed in this revised version. The text now flows much more smoothly, and the phrasing is now cleared. I congratulate the authors on this improvement, as it greatly enhances the manuscript's readability. Regarding the biological interpretation of the data, I see a noticeable improvement in this aspect as well. While challenges remain in making biological sense of all the generated data, I appreciate the authors' efforts to provide additional context and their arguments highlighting the inherent difficulties in this area. These revisions demonstrate thoughtful engagement with the feedback provided.
However, I still feel that the quality of the images remains a significant issue. The current visualizations do not do justice to the quality of the work and, unfortunately, detract from the overall impact of the manuscript. The use of print screens from analysis tools not only reflects a lack of effort in improving the presentation but also raises potential copyright concerns, which should be addressed by the journal. While the authors reference other studies to justify the use of certain types of visualizations (e.g., bar charts, Venn diagrams, and network charts), I believe the quality of these visualizations in their manuscript falls short in comparison. While the types of graphs are similar, they do not approach the same standard in terms of clarity, detail, or overall presentation. As this is a point of disagreement between the authors and myself, I will leave it to the editor to assess and make a final decision.
Author response
Thank your contribution and thorough review and editing suggestions provided to us to improve this work. The manuscript review process has shaped the way the data is presented and articulated in this manuscript (for both Figures and Tables) as well as major grammatical issues that were severely criticised by the Reviewer. We wish to state that for this manuscript, we explored all the available tools (Venn diagrams, Heat maps, several enrichment analysis including gene networks, secondary structures of non-coding RNAs and their respective mRNA targets) to show that genetic heterogeneity within clinical strains of M. tuberculosis is not a random occurrence but can be linked to both coding and non-coding RNAs that have been previously unexplored (not to the depth that have been presented in this manuscript for non-coding RNAs), which has been limited to very few small RNAs with known biological functions. Lastly, we wish to make an emphasis on the images used in this manuscript. We used several Bioinformatics tools for visualization, and these are not print screen images. These images were downloaded from open-source databases and softwares which are cited following the developers instructions. For specific details, please see our responses below.
Reviewer comment
Line 50: There is an extra comma after the word "complex." Please consider removing it.
Author response
Line 51: the space after complex has been removed.
Reviewer comment
Line 111: One extra space is found in this line. Kindly remove it for consistency.
Author response
Line 111: The space after “high” has been removed.
Reviewer comment
Line 123: One extra space is found in this line. Please correct this formatting issue.
Author response
Line 119 – 122: Upon accepting the changes made in word document, no spaces were detected in this line. It could’ve been due to edits made prior to accepting text insertions and deletions.
Reviewer comment
Line 171: One extra space is found in this line. It would be great to fix this to ensure uniformity.
Author response
Line 161 – Line 165: Upon accepting the changes made in word document, no spaces were detected in this line. It could’ve been due to edits made prior to accepting text insertions and deletions.
Reviewer comment
Table 2: There seems to be multiple font types or sizes used. It would enhance the presentation if the same font type and size were used consistently throughout the table.
Author response
Table 2 has been reformatted to Palatino Linotype (font 10) which is consistent to Table 1 formatting.
Reviewer comment
Line 298: There is one extra space, and a sentence starts without a capital letter. Please correct this for proper grammar and formatting.
Author response
Line 297: Extra space has been deleted and the sentence now start with a capital letter.
Reviewer comment
Figure 2: This figure contains print screens of an online tool. I recommend verifying the copyright for these images. Additionally, the meaning of FDR should be included in the figure legend for clarity.
Author response
Figure 2 is not a print screen image. The STRING database is a Bioinformatics tools that collate the list of genes provided and create interactions within protein products of these genes based on their locations, predicted and known interactions with each other based on genome annotation of M. tuberculosis. Once the gene network is created, this can be visualized as an image (Figure 2 and Figure 3 in our manuscript). Bioinformatics databases and softwares are usually open for all academic purposes (unless it a closed tools that require payment) as long as they remain cited (STRING is cited in Line 553 of the manuscript) as instructed in this link under the tab, “How to cite” https://version-12-0.string-db.org/cgi/help?sessionId=bzNLttVvwTno. There is no copyright issues associated with images created within Bioinformatics databases (and sometimes softwares) since they were created to provide visualization of specific data (in this case, protein networks) during Bioinformatics analysis. Lastly, once the network graph is created, it is exported as a figure (please see this link, specifically for Figure 2; https://version-12-0.string-db.org/cgi/network?taskId=bHE9kXf6atjG&sessionId=bzNLttVvwTno). We did not print screen these images (there is an export option to download the created image as a bitmap, PNG, etc). If the link provided does not open, you may need to create an account before you can access data within STRING so that we can verify that these images were not print screen images. As of November 2020, there have been >3 million publications that have used the STRING database as an interactive gene network platform which is cited as advised by developers.
Reviewer comment
Figure 3: This figure also contains print screens of an online tool. Please verify the copyright status. It would be helpful to include an explanation of the FDR in the figure legend.
Author response
Figure 3: The FDR (False discovery rate) explanation has been provided in the figure legend. Figure 3 is not a print screen image as explained previously for Figure 2. Here is the link for Figure 3A to access and download the gene network created within this database: https://version-12-0.string-db.org/cgi/network?taskId=bac4Fl6R9SmC&sessionId=bzNLttVvwTno. This figure was exported based on protein product of the genes that interact with Transcript 354 of the Beijing strain.
Reviewer comment
Figure 4: This figure contains print screens of an online tool. Kindly verify the copyright for these images.
Author response
Figure 4 is not a print screen of an online tool. Figure 4 A was created within Venny database (which was cited in Line 554) as per the developers instructions. To date, more than 2000 publications have used this Bioinformatics tools. Figure 4B was created using MeV software (cited in Line 554), Figure 4C (RNA Fold Webserver) cited in Line 549 and Figure 4D was created within TargetRNA3 database is cited in Line 550. All tools used have been properly acknowledged and cited as per previous publications that exploited these tools, hence, there are no copyright issues associated with them.
Reviewer comment
Regarding the STRING print screens: What kind of biological insights can one derive from a network where most genes are identified only by their locus, and functions are not attributed? One of the references the authors mention provides a more informative network with annotated functions highlighted within the network, offering clearer biological context. I believe this approach would strengthen the manuscript.
Author response
Figure 2 and Figure 3 shown gene networks that are more likely to be impacted by the non-coding RNAs (Transcript 161 for the KZN and Transcript 354 for the Beijing strain). The functional categories for each of these figures (Figure 2B and Figure 3C) were included (with a q value of < 0.05 denoted by their respective FDR) to show us which interactions will most likely be affected by high expression of the ncRNAs. Lastly, once all ncRNAs are identified within clinical strains of M. tuberculosis, their interactions with their respective targets will be known, which can (future work) be presented with the ncRNA within the gene network with it respective probability of binding.
Reviewer comment
For all figures and tables where "ncRNA structure color coding" is used, I would suggest explaining in the legend what the color coding represents, such as whether it encodes probability or confidence levels for the proposed structure. This clarification will help readers better interpret the visualized data.
Author response
Table 4, Table 6 and Table 8 footnote has been edited as follows: *: ncRNA structure color coding. Blue indicates lowest probability of binding while red denote high probability of binding.
Figure 4 colour coding has been edited as follows: : Blue indicates lowest probability of binding while red denote high probability of binding.
We trust that we have adequately responded to the Reviewers’ comments and queries.
Thank you for your consideration.
Kind Regards
Dr. Nontobeko Mvubu (PhD)
Senior Lecturer/Researcher
719 Umbilo Road
Doris Duke Medical Research Institute
School of Laboratory Medicine and Medical Sciences
Discipline of Medical Microbiology
College of Health Sciences
University of KwaZulu-Natal
Durban, 4000
South Africa
Email: mvubun@ukzn.ac.za
Tel: +27312601934
Round 3
Reviewer 2 Report
Comments and Suggestions for Authors
Dear authors,
Thank you for the effort you have put into revising your manuscript through this review process. I appreciate the work you have done to address the comments provided in earlier rounds.
At this stage, my only remaining concern is with the quality of the visualizations. Unfortunately, I feel that these figures still do not meet the standards required for publication. Despite the feedback provided in earlier rounds, there have been no significant improvements in this area.
As I believe I have provided all the guidance I can on this matter, I have no further comments to contribute. I trust that the editor will carefully evaluate this aspect and make the final decision regarding the manuscript.
I wish you all the best with your work and hope my feedback has been helpful in improving your manuscript overall.
Best regards
Author Response
Reviewer comment
Dear authors,
Thank you for the effort you have put into revising your manuscript through this review process. I appreciate the work you have done to address the comments provided in earlier rounds.
At this stage, my only remaining concern is with the quality of the visualizations. Unfortunately, I feel that these figures still do not meet the standards required for publication. Despite the feedback provided in earlier rounds, there have been no significant improvements in this area.
As I believe I have provided all the guidance I can on this matter, I have no further comments to contribute. I trust that the editor will carefully evaluate this aspect and make the final decision regarding the manuscript.
I wish you all the best with your work and hope my feedback has been helpful in improving your manuscript overall.
Best regards
Author Response
We would like to take this opportunity and thank the Reviewer for all the comments and contributions to improve this work at a scientific as well as the technical level. Please find below the measures taken to improve figure quality for better visualization and communication:
Figure 1A has been changed from blue and yellow colour combination to plain black and white.
Figure 1B – 1F: The exes have been bolded and the font was increased.
Table 4, Table 6 and Table 8: The sRNA figures within Table 4, Table 6 and Table 8 have been changed from colour to black and white (Since these figures are reformatted to fit into the Tables, original images are available upon request).
Figure 2A and 2B were downloaded as bitmap with better saturation and visibility of the interacting proteins within the network and functional enrichment, respectively.
Figure 3A and 3B were downloaded as bitmap with better saturation and visibility of the interacting proteins within the network.
Figure 4A has been changed from blue and yellow colour combination to plain black and white.
Figure 4C has also been changed to back and white instead of colours to improve readability of the sequences within the predicted sRNA structure.
Figure 4D saturation has been increased to improve the quality of the image.
We trust that we have adequately responded to the Reviewers’ comments and queries.
Thank you for your consideration.
Kind Regards
Dr. Nontobeko Mvubu (PhD)
Senior Lecturer/Researcher
719 Umbilo Road
Doris Duke Medical Research Institute
School of Laboratory Medicine and Medical Sciences
Discipline of Medical Microbiology
College of Health Sciences
University of KwaZulu-Natal
Durban, 4000
South Africa
Email: mvubun@ukzn.ac.za
Tel: +27312601934